# Research on Agricultural and Rural Public Governance and Sustainable Development: Evidence from 2350 Data

**Tingting Huang** [1,2] **and Qinghua Huang** [3,*]

1    School of Public Policy and Management, Guangxi University, Nanning 530004, China; sherohtt@foxmail.com
2    School of Economics and Management, Tongren University, Tongren 554300, China
3    School of Economics and Management, Southwest University, Chongqing 400715, China
*    Correspondence: hqh@swu.edu.cn

**Abstract:** Sustainable agriculture and good governance are part of the UN Sustainable Development Goals (SDGs), which have attracted great attention from all nations around the world. A scientific metrological and knowledge map analysis was conducted on the spatial–temporal evolution, collaboration network, research hotspots, cluster labels, frontier detection, and evolution path of 2350 pieces of data in this paper. The main results show that research hotspots such as sustainable development, rural development, agriculture, and others have influenced the development of the entire research process and have evolved into larger topic cluster groups such as ecosystem service, sustainable agriculture, land consolidation, and agricultural intensification. Research frontiers such as agriculture, integrated systems, smallholder systems, rural sustainable development, and land consolidation play key roles. Based on the findings, it is necessary to focus on the UN 2030 SDGs, combine the countries' regional development needs and reality, and further clarify and refine the topics that need to be studied and the problems that need to be solved. More scientific demonstration and more feasible measures should be adopted to jointly deal with and enhance awareness of the current problems and practical challenges and further promote practical development by cohering academic consensus and expanding and innovating the governance models from the comprehensive dimensions of economy, politics, society, culture, and ecological environment so as to achieve good agricultural and rural governance.

**Keywords:** agricultural and rural; sustainable development; public governance; scientific metrological analysis; knowledge map

## 1. Introduction

Promoting sustainable agriculture and good governance both belong to the SDGs. Agriculture underpins rural livelihoods and rural economies; villages are homes for rural residents and rural economies, with a vital role in promoting the achievement of the Sustainable Development Goals for agricultural and rural systems. Because of such, the issue of agricultural and rural public governance and sustainable development has become an important topic that has attracted great attention from all nations around the world. Looking far and wide at existing theories and practices, agricultural and rural sustainable development is related to the common prosperity of farmers and rural areas [1]. Agricultural and rural public governance reflect the governance capacity and governance level of the state for agricultural and rural areas. Obviously, these important topics have also attracted widespread attention and in-depth discussion from scholars. On the whole, public governance and sustainable development in agricultural and rural areas have various economic, social, and environmental benefits [2] and embody the following goal requirements at the macro level, such as an overall good economic and income situation for farmers, good ecological conditions, rich biodiversity, relatively well-established governance structures, the identities of the people within the rural region, and comparatively low social

discrepancy [3]. At the same time, scholars have combed out a number of conditions that contribute to public governance and sustainable development in agricultural and rural areas. The following aspects are included: reducing deforestation, carrying out the intensification management of agricultural regions [4], transforming the utilization mode of cropland in rural areas [5], putting into effect the grassroots anti-pesticide mobilization [6], developing agricultural extension activities, supporting rural credit, improving agricultural mechanization, expanding agricultural and rural marketing [7], developing demonstration farms [8–13], cultivating a community for rural environmental governance [14], and increasing agricultural production efficiency [15].

However, there are also some practical and empirical results that reveal the reasons or factors that hinder public governance and sustainable development in agriculture and rural areas, for instance, agricultural land fragmentation [16], changes in climate, technology, policy, and market prices [17], a lack of leadership and overall planning for the sustainable development of resources and environment [18], weak rural education, inadequate labor supply, agricultural extension services that are not yet universal, insufficient social capital, risk mitigation attitudes that are not optimistic enough, less farming experience, and restrictions due to soil conditions [2]. In conclusion, there are many factors, which can be summed up as follows: agricultural economy, agricultural productivity, farm size, market access, agroecological potential, agricultural product supply chain, rural industrial, work environment, living conditions, infrastructure, public services, public involvement, rural culture, government-related departments, educational resources, health and welfare, social governance, natural, physical, environmental, financial, and social capital, as well as corporate social responsibility [19–24].

For all these reasons, it is necessary to improve the economic, social, and environmental influence of agricultural and rural areas as a whole [25]. Government and non-governmental organizations must improve their rural and agricultural development policies [26], define policies that are socially and environmentally acceptable and geared towards tackling complex challenges [27], and advance the development of plans and strategies for the sustainable development of villages [28]. Priority should be given to the construction of transportation infrastructure, regulation of farmland transfer, industrial integration, promotion of rural entrepreneurship and land consolidation [29]; improving public investments in infrastructures, human capital, and technology in agriculture and rural area to enhance the competitiveness [30]; strengthening more effective public forest governance [4]; developing multi-talent rural education and integrating first, second, and third industries [31–33]; facilitating bridging the technical and associative potential of agroecological production [34]; and increasing farmers' income, forming a more complete agricultural product supply chain, and highlighting the agricultural brand effect [35,36]. Simultaneously, more attention needs to be paid to the reconstruction of governance structures and governance models in rural areas [37]. Explorations toward developing a geoscientific approach to public governance and sustainable development in agriculture and rural areas are important [38], as is an in-depth analysis of the internal mechanisms of the evolution of agricultural production patterns at different phases [39] and giving impetus to multifunctional rural development [40]. From this, it can be seen that achieving public governance and sustainable development in agricultural and rural areas and the improvement of agricultural economic environments and rural development environments should be considered [21].

To summarize, there is no denying the fact that the academic literature provides a noteworthy reference for decision makers in their follow-up public governance and sustainable development planning in agriculture and rural areas. However, objectively speaking, although there is a large amount of existing research on public governance and sustainable development in agricultural and rural areas, it rarely focuses on dynamic progress, hot spot analysis, frontier detection, evolution logic, and trend outlook. The scientific metrological analysis of previous research results is even more rare, which is

not conducive to other scholars or readers recognizing the importance, authority, and representativeness of research results from the massive amount of literature data.

In light of this, the purpose of this paper is to focus on taking agricultural and rural public governance and sustainable development as the research theme. Based on the WoS core collection database and CiteSpace metrological analysis software, this study provides an in-depth examination of the spatial–temporal evolution, cutting-edge map, and logical evolution of research on public governance and sustainable development in agricultural and rural areas. Firstly, it can contribute to revealing and reflecting the status and progress of topics and hot spots, as well as the mainstream and fronts, and the trends and vistas in its research field from a systematic, comprehensive, and whole perspective for research on public governance and sustainable development in agricultural and rural areas. Secondly, it can contribute to fully grasping the logical context of the development history and the frontier trends of hot knowledge for research on public governance and sustainable development in agricultural and rural areas. Thirdly, it can contribute to providing academic perspectives and research foundations for other scholars or decision makers to engage in related research on agricultural and rural public governance and sustainable development.

The rest of the paper is structured as follows: Section 2 provides the research method and data sources. Section 3 describes the basic situation. Section 4 details the results and discussions. Section 5 summarizes the conclusions of this paper.

## 2. Research Method and Data Sources

### 2.1. Conceptual Model

This paper draws on the concept model of CiteSpace pioneered by Prof. Chaomei Chen [41,42]; this model creatively integrates the methods of citation analysis (diachronic) and co-citation analysis (structural), creates the mapping from knowledge base to research frontier, and highlights the discipline basis of the citation network map and the technical basis of the information space map [42,43]. With the help of version V.6.2.R1 of the CiteSpace specialized scientific metrological analysis software with relatively complete functions, this paper expects to analyze the basic situation, research hotspots, frontier detection, and evolution path of agricultural and rural public governance and sustainable development, in order to find the potential associations contained in the research on agricultural rural public governance and sustainable development, and characterize a series of visual knowledge maps to reveal and reflect the multivariate, time-sharing, and dynamic spatial information, as well as its network relationships and mutual influences, and then interrogate the temporal and spatial variations, frontier probing, and evolution paths in the field of agricultural rural public governance and sustainable development. The conceptual model of CiteSpace is shown in Figure 1.

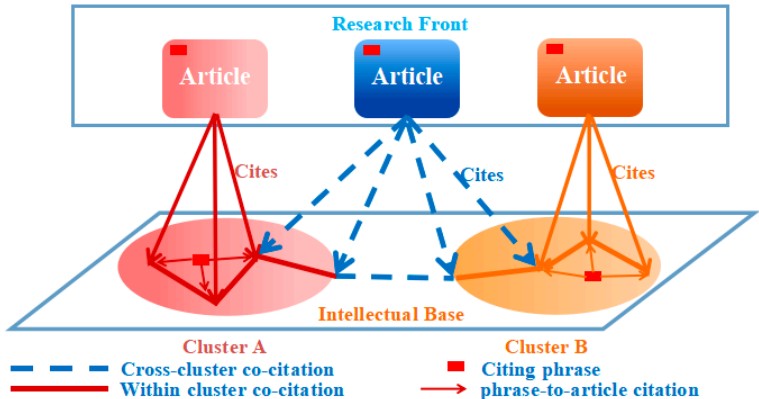

**Figure 1.** The conceptual model of CiteSpace.

*2.2. Research Methods*

In this paper, according to the research goal and conceptual model, an application model of CiteSpace for research on public governance and sustainable development in agricultural and rural areas was created. The application model of CiteSpace on agricultural and rural public governance and sustainable development is depicted in Figure 2. Furthermore, this paper attempts to answer the following questions:

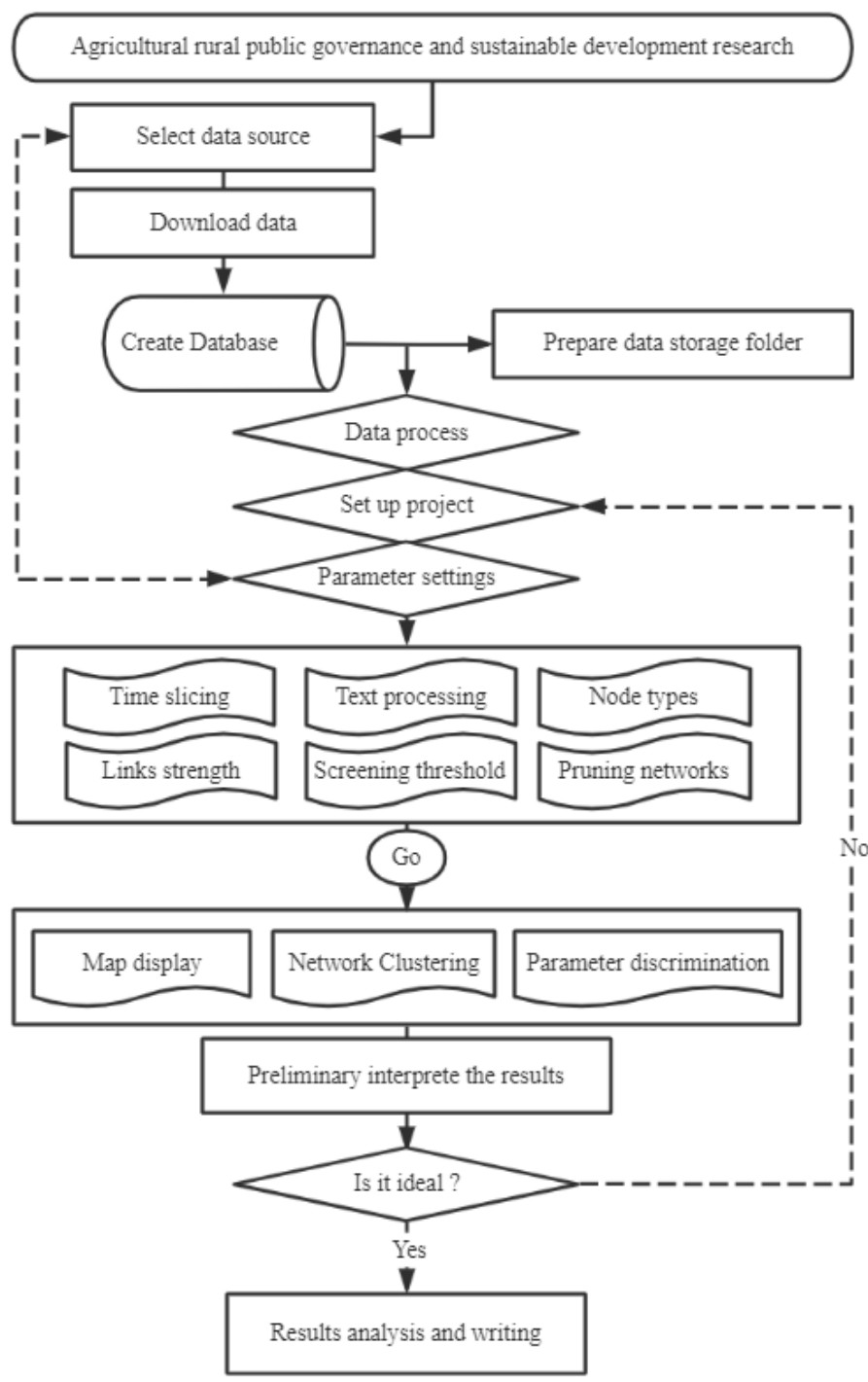

**Figure 2.** The application model of CiteSpace on agricultural and rural public governance and sustainable development.

(1) When did the field of agricultural and rural public governance and sustainable development start to be studied? Where is the research stronger? What are well-known scholars paying attention to? What is the cooperation network in this field?

(2) What research hotspots have emerged in the field of agricultural and rural public governance and sustainable development? What research topic clusters have been identified? What research frontiers are presented?

(3) How has the knowledge base and research paradigm in the field of agricultural and rural public governance and sustainable development evolved? In which years did a cluster appear? In which years did the research results of a certain cluster begin to increase or decrease? In which years did iconic literature appear and affect the overall trend of clustering?

Taking the above analyses together, combined with the practical aspects of the research content of this paper, the following theoretical hypotheses are proposed:

**Hypothesis 1.** *Scholars from different countries or regions have a certain degree of cooperation in the research field of agricultural and rural public governance and sustainable development, but the network relationship of academic cooperation is not very strong.*

**Hypothesis 2.** *The research hotspots in the research field of agricultural and rural public governance and sustainable development embody the SDGs on the macro level, but the specific research hotspots in different periods are uncertain.*

**Hypothesis 3.** *The evolution path of the knowledge base and research paradigm in the research field of agricultural and rural public governance and sustainable development are uncertain.*

Among them, the important calculation formulas involved are introduced as follows.

### 2.2.1. Time Slicing

CiteSpace provides entropy as a macroscopic indicator for measuring the orderliness and disorder of the network at different time periods [42]. This paper sets up one time slice per year. Entropy is defined as follows:

$$E = -\sum_{i=1}^{n} p(x_i) log\, p(x_i) \tag{1}$$

According to Equation (1), where $E$ is the entropy, $P_i$ is the probability of node $i$ appearing in the network.

### 2.2.2. Links Strength

The links parameter is an important basis for calculating the links strength of network nodes [42,43]. In this paper, the Cosine algorithm is selected, and the calculation method is as follows:

$$Cosine(x,y) = \frac{XY}{\|X\|\|Y\|} = \frac{C_x C_y}{\sqrt{\left(\sum_{i=1} C_{x_i}\right)^2 \left(\sum_{i=1} C_{y_i}\right)^2}} \tag{2}$$

In Equation (2), $C_x$ and $C_y$ stands for the co-occurrence times of $x$ and $y$, respectively; $\left(\sum_{i=1} C_{x_i}\right)^2$ and $\left(\sum_{i=1} C_{y_i}\right)^2$ stand for the frequency of $C_x$ and $C_y$, respectively.

### 2.2.3. Selection Criteria

In this paper, the extraction method of knowledge units is mainly based on the modified $g$-index for ranking [42,43]. The formula for the g-index is as follows:

$$g = \frac{1}{g} \sum_{i=1}^{g} (k \cdot C_i) \tag{3}$$

Among them, *g* indicates the *g*-index and *k* indicates the scale factor. According to the knowledge unit network of the sample data in this paper and adjusting appropriately according to the suggestions from related scholars, *k* is set to 20 in this paper. At the same time, default values were chosen for the parameters of Top N and Top N%.

### 2.2.4. Betweenness Centrality

Betweenness centrality is an indicator that measures the importance of nodes in a network and is quite critical; it measures and reflects the importance of the literature [43]. In general, the literature with high betweenness centrality is the key link connecting two different research fields, and when the literature is highlighted by the purple ring, it means that this type of literature is very important and its betweenness centrality is usually greater than 0.1. The calculation formula for betweenness centrality is as follows:

$$BC_i = \sum_{n_a \neq i \neq n_b} \frac{n_{n_a n_b}^i}{g_{n_a n_b}} \tag{4}$$

In Equation (4), where $g_{n_a n_b}$ is the number of shortest paths from node $n_a$ to node $n_b$, $n_{n_a n_b}^i$ is the number of shortest paths passing through node $i$ among the $g_{n_a n_b}$ shortest paths from node $n_a$ to node $n_b$. From the perspective of information transmission, the higher the betweenness centrality, the more prominently it reflects the importance of nodes.

### 2.2.5. Sigma Index

The sigma index is an important indicator of the novelty of a measure node composed of two indicator composites according to the importance of the node in the network structure (betweenness centrality) and the importance of the node in the time course (burstness) [43]. The calculation method is as follows:

$$Sigma = (centrality + 1)^{burstness} \tag{5}$$

### 2.2.6. Modularity and Silhouette

Modularity is an important indicator for evaluating the effectiveness of community identification [38,40]. The calculation formula for modularity is as follows:

$$Modularity \ Q = \sum_i \left( l_{ii} - p_i^2 \right) \tag{6}$$

According to Equation (6), $i$ denotes the number of communities that have been divided, $l_{ii}$ is the proportion of internal links within the community in relation to all links in the full diagram, and $p_i$ is the proportion of links related to community $i$ in relation to all links in the full diagram. When the clustering effect is better, there will be more links within the community. By analogy, the larger the $l_{ii}$, the larger the modularity $Q$. Generally, the range of modularity $Q$ is $[0, 1)$, when $Q > 0.3$ (empirical value), which means that the community structure divided is significant [42,43].

Meanwhile, it should be noted that silhouette is also an important parameter indicator for evaluating the clustering effect of a community [42,43]. The calculation formula for silhouette is as follows:

$$Silhouette \ S = 1 - \frac{\alpha_i}{\beta_i} \tag{7}$$

Among them, $\alpha_i$ is the average distance between node $i$ and the other nodes in the cluster where it is located, and $\beta_i$ is the average distance between node $i$ and the other nodes in the cluster where the nearest node $i$ is located.

### 2.3. Data Collection

In order to comprehensively consider the uniformity, consistency, scale, balance, and authority of data sources, and provide an overall consideration of the recall and

accuracy of research samples, the original data in this paper came from the core collection database of Web of Science, and the conditions of the literature retrieval were set as follows: (TS = (agricultural and rural public governance) OR TS = (agricultural and rural sustainable development)) AND (DT == ("ARTICLE"). A total of 2350 pieces of literature data were obtained in the time frame from 1996 to 2023. The date of retrieving data was 24 March 2023.

## 3. Basic Situation Analysis

### 3.1. Trends in the Number of Published Papers

Figure 3 depicts the trend of published papers on agricultural and rural public governance and sustainable development, which helps in understanding the publication dynamics of research on public governance and sustainable development in agricultural and rural areas. From the overall characteristics, since 1996, the core literature on agricultural and rural public governance and sustainable development research has appeared, with the fluctuation in the number of publications in the literature on agricultural and rural public governance and sustainable development being very flat between 1996 and 2015, with no more than 70 publications in the highest annual volume. However, it is surprising that from 2016 to 2022, the number of literature publications showed a rapid linear increase. During this period, the lowest annual publication volume was 81 in 2016 and reached the highest annual publication volume of 420 in 2022. This means that after 2015, the number of publications in the research field of agricultural and rural public governance and sustainable development shows a steep growth slope, which has a great relationship with the UN Summit on Sustainable Development in 2015, which adopted 17 Sustainable Development Goals with the aim of turning to the path of sustainable development, and has also caused in-depth research and extensive discussion on this theme in the academic community.

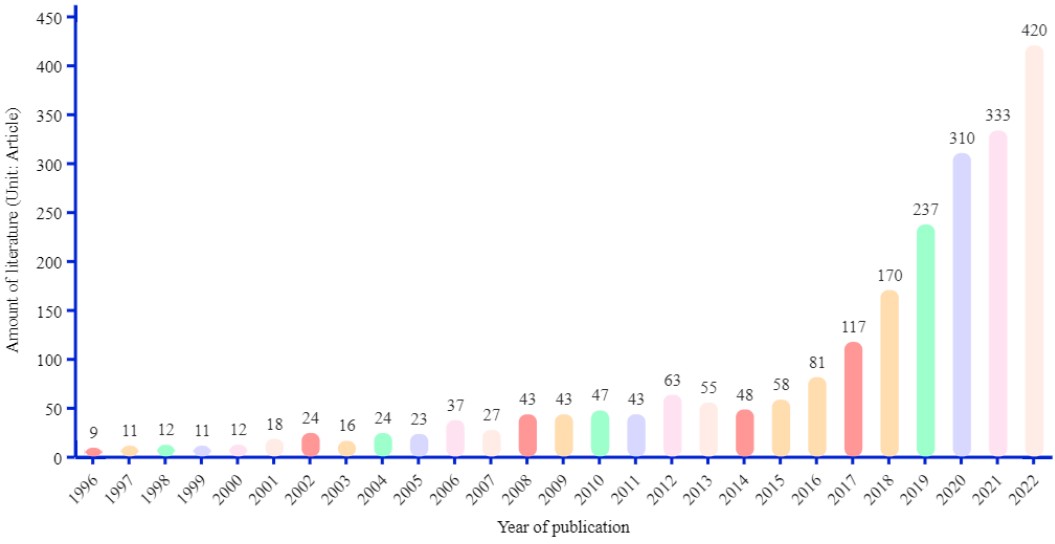

**Figure 3.** Trend of published papers on agricultural and rural public governance and sustainable development.

### 3.2. Author Cooperation Networks

The network analysis of author collaboration can embody the importance indicators and network relationship attributes of each author in the network in the field of agricultural and rural public governance and sustainable development. Figure 4 clearly characterizes the author collaboration network map on agricultural and rural public governance and sustainable development. Viewed from the size of the nodes, the size of the nodes reveals the high and low frequency of publications in the literature. The top-ranked scholars in terms of number of publications are as follows: Li YR; Khan N; Zhang SM; Shmatkovska T; Dziamulych M; Abbas A; etc. Viewed from the network structure of the nodes, the map of the author cooperative network consists of 805 nodes and 502 lines. The density of the

co-occurrence network is 0.0016, which indicates that the cooperation intensity needs to be strengthened.

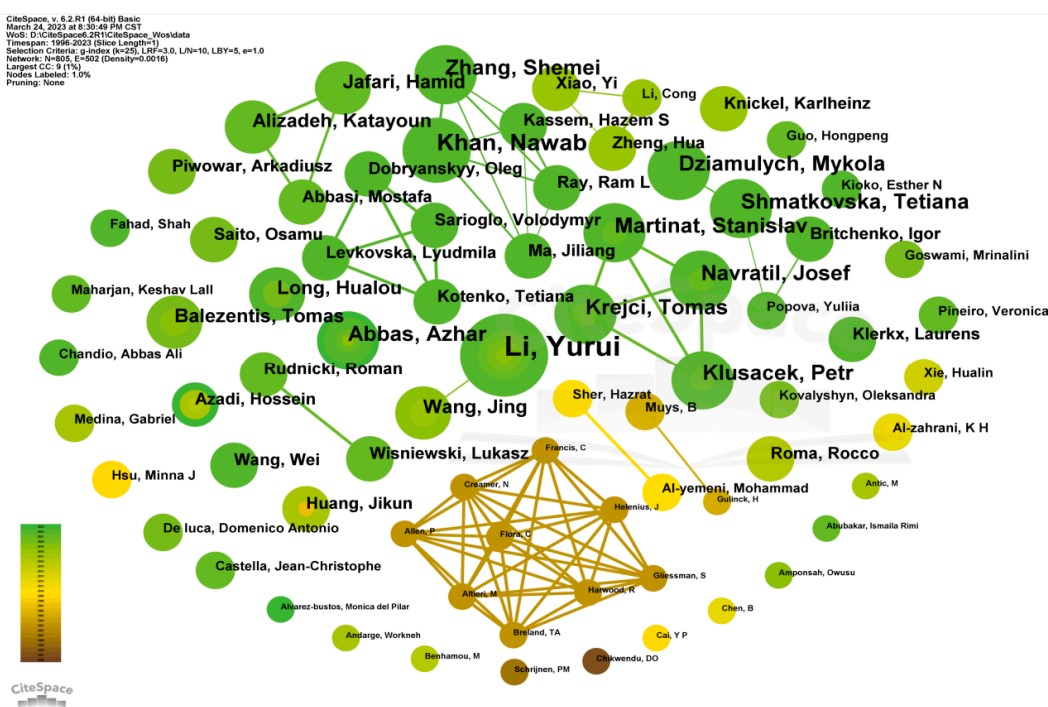

**Figure 4.** Author collaboration network map.

Table 1 sorts the 10 authors with the highest number of times cited and their literature distribution. In the research field of public governance and sustainable development in agricultural and rural areas, a larger group of contributing scholars and more prominent results were formed, which were cited with a higher frequency, whether in the WoS core database or all databases. For example, Bryan BA published "China's response to a national land-system sustainability emergency" in 2018 to discuss the issue of national land system sustainability and was cited as frequently as 597 times in total, ranking first among the highly cited authors. Secondly, Nobre CA discussed the topic of a novel sustainable development paradigm in the "Land-use and climate change risks in the Amazon and the need of a novel sustainable development paradigm" published in 2016, and is the second most highly cited author with a total citation frequency of 375. Ranked third is Wu YY, discussing the subject of the overuse of agricultural chemicals in the 2018 publication of "Policy distortions, farm size, and the overuse of agricultural chemicals in China", with a total citation frequency of 350. It can be seen that these highly cited research results have a strong influence on and dissemination force in the academic community, but we also find that scholars with a higher volume of publications are not necessarily highly cited authors.

### 3.3. Institution Cooperation Networks

The network analysis of institutional cooperation can reflect the layout of research forces in the field of agricultural and rural public governance and sustainable development. Figure 5 vividly draws a research institution collaboration network map on agricultural and rural public governance and sustainable development. Taken as a whole, the map of the research institution cooperative network consists of 566 nodes and 1238 lines, and the density of the co-occurrence network is 0.0077. Meanwhile, three purple annual circles have emerged in the map, which means that the following three research institutions appeared more frequently in the corresponding years, namely the Chinese Academy of Sciences, Wageningen University & Research, and CGIAR. Comparatively speaking, a certain cooperative network relationship has been formed between research institutions,

which has laid an important foundation for exploring the issues of agricultural and rural public governance and sustainable development.

**Table 1.** The 10 authors with the highest number of times cited and their literature distribution.

| Rank | Author | Article Title | Times Cited | | Publication Year |
|---|---|---|---|---|---|
| | | | WoS Core | All Databases | |
| 1 | Bryan BA | China's response to a national land-system sustainability emergency | 535 | 597 | 2018 |
| 2 | Nobre CA | Land-use and climate change risks in the Amazon and the need of a novel sustainable development paradigm | 367 | 375 | 2016 |
| 3 | Wu YY | Policy distortions, farm size, and the overuse of agricultural chemicals in China | 321 | 350 | 2018 |
| 4 | Kassie M | Adoption of interrelated sustainable agricultural practices in smallholder systems: Evidence from rural Tanzania | 329 | 342 | 2013 |
| 5 | Teklewold H | Adoption of Multiple Sustainable Agricultural Practices in Rural Ethiopia | 311 | 316 | 2013 |
| 6 | Wang J | Land-use changes and policy dimension driving forces in China: Present, trend and future | 240 | 269 | 2012 |
| 7 | Jayne TS | Land pressures, the evolution of farming systems, and development strategies in Africa: A synthesis | 261 | 266 | 2014 |
| 8 | Dearing J | Safe and just operating spaces for regional social-ecological systems | 250 | 262 | 2014 |
| 9 | Srbinovska M | Environmental parameters monitoring in precision agriculture using wireless sensor networks | 250 | 260 | 2015 |
| 10 | Sumane S | Local and farmers' knowledge matters! How integrating informal and formal knowledge enhances sustainable and resilient agriculture | 246 | 253 | 2018 |

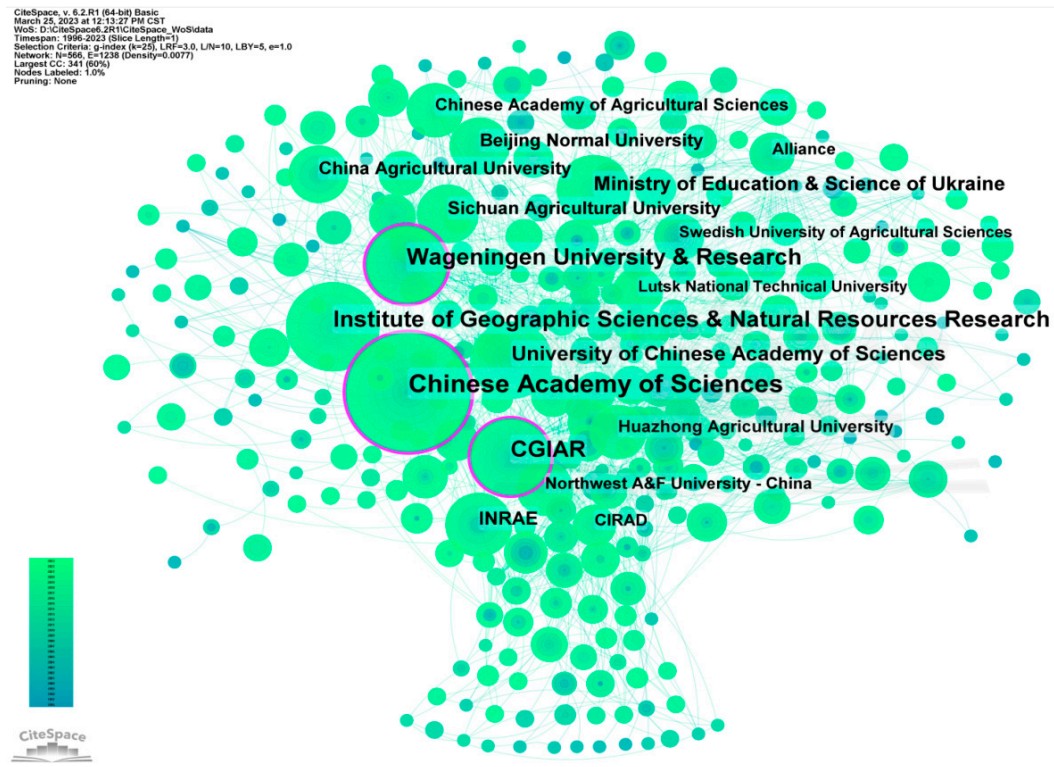

**Figure 5.** Research institution collaboration network map.

To provide further detail, Table 2 arranges the 20 institutions with the highest total number and centrality in agricultural and rural public governance and sustainable development research. The centrality of these research institutes is more than 0.01, which indicates that these research institutes occupy a very important academic position in the corresponding years, where their role is prominent and their impact is significant.

**Table 2.** The 20 institutions with the highest total number and centrality.

| Rank | Institutions | Year | Count | Centrality |
|---|---|---|---|---|
| 1 | Chinese Academy of Sciences | 2003 | 107 | 0.14 |
| 2 | Wageningen University & Research | 1998 | 69 | 0.14 |
| 3 | CGIAR | 2000 | 70 | 0.12 |
| 4 | Chinese Academy of Agricultural Sciences | 2010 | 22 | 0.08 |
| 5 | Swedish University of Agricultural Sciences | 2003 | 15 | 0.08 |
| 6 | Poznan University of Life Sciences | 2020 | 9 | 0.05 |
| 7 | Ministry of Education & Science of Ukraine | 2018 | 35 | 0.04 |
| 8 | Alliance | 2010 | 18 | 0.04 |
| 9 | Leibniz Zentrum fur Agrarlandschaftsforschung | 2009 | 14 | 0.04 |
| 10 | Ghent University | 2009 | 11 | 0.04 |
| 11 | Bangor University | 2000 | 6 | 0.04 |
| 12 | Institute of Geographic Sciences & Natural Resources Research | 2006 | 68 | 0.03 |
| 13 | INRAE | 1997 | 33 | 0.03 |
| 14 | Beijing Normal University | 2009 | 25 | 0.03 |
| 15 | Huazhong Agricultural University | 2018 | 21 | 0.03 |
| 16 | CIRAD | 2006 | 18 | 0.03 |
| 17 | Empresa Brasileira de Pesquisa Agropecuaria | 1999 | 14 | 0.03 |
| 18 | University of California System | 1996 | 13 | 0.03 |
| 19 | Indian Council of Agricultural Research | 2006 | 12 | 0.03 |
| 20 | Universidad Politecnica de Madrid | 2010 | 9 | 0.03 |

*3.4. Country/Region Cooperation Networks*

The network analysis of country/region cooperation can outline the distribution of cooperation and the strength of cooperation relationships among different countries in the field of agricultural and rural public governance and sustainable development. Figure 6 plainly portrays the country/region cooperation network map for agricultural and rural public governance and sustainable development. From the information in the map's structure, the map of the country/region cooperative network consists of 136 nodes and 1175 lines, and the density of the co-occurrence network is 0.128. It reflects that the cooperative relationship between countries has a relatively high strength. In particular, since 1996, research in the field of agricultural and rural public governance and sustainable development has occurred at a relatively high frequency in the following countries/regions: People's R. China, USA, England, Italy, Germany, Netherlands, Australia, India, Spain, France, and so on.

Table 3 lists the 20 countries or regions with the highest total number and centrality in agricultural and rural public governance and sustainable development. The 10 countries/regions with the highest node centrality are England, USA, Germany, Australia, Italy, Canada, Slovakia, Netherlands, Peoples R. China, and India, respectively, which also illustrates that the above countries/regions have higher impacts on agricultural and rural public governance and sustainable development research.

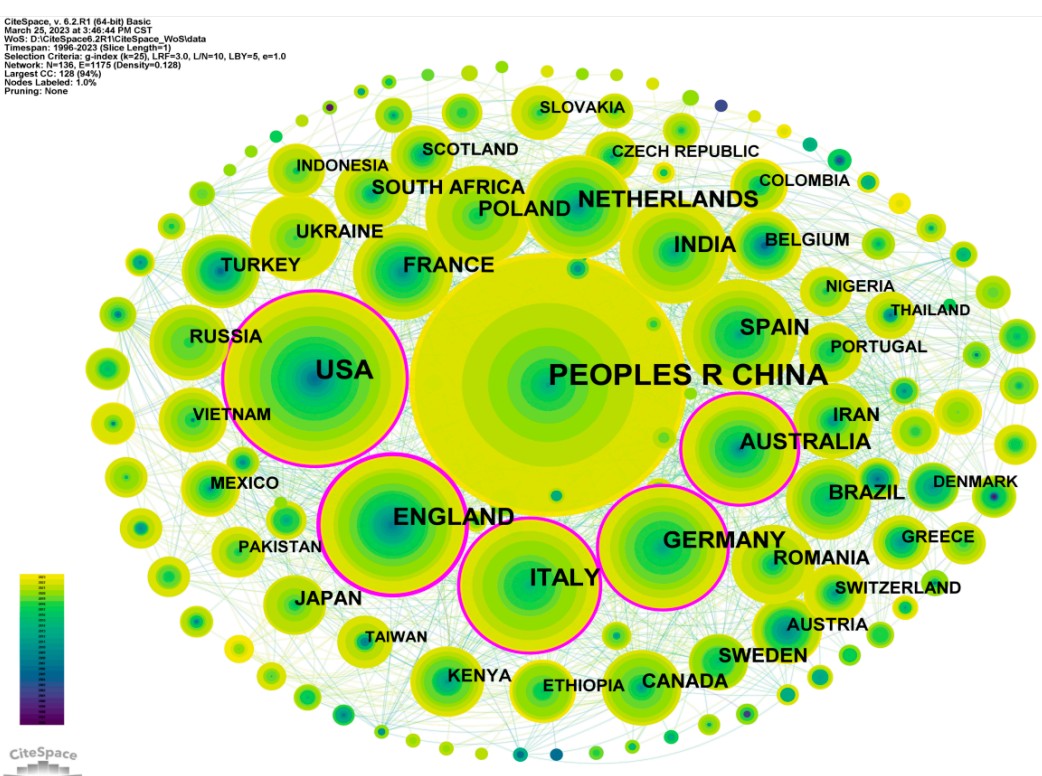

**Figure 6.** Country/region cooperation network map.

**Table 3.** The 20 countries or regions with the highest total number and centrality.

| Rank | Countries/Regions | Year | Count | Centrality |
|------|-------------------|------|-------|------------|
| 1 | England | 1997 | 162 | 0.21 |
| 2 | USA | 1996 | 261 | 0.18 |
| 3 | Germany | 2001 | 129 | 0.16 |
| 4 | Australia | 1998 | 107 | 0.16 |
| 5 | Italy | 2002 | 145 | 0.11 |
| 6 | Canada | 1996 | 60 | 0.10 |
| 7 | Slovakia | 2009 | 24 | 0.10 |
| 8 | Netherlands | 1998 | 116 | 0.09 |
| 9 | Peoples R. China | 1998 | 543 | 0.07 |
| 10 | India | 1997 | 106 | 0.07 |
| 11 | Spain | 2008 | 100 | 0.07 |
| 12 | France | 1997 | 83 | 0.06 |
| 13 | Russia | 1997 | 42 | 0.06 |
| 14 | Kenya | 1999 | 40 | 0.05 |
| 15 | Scotland | 2005 | 33 | 0.05 |
| 16 | Japan | 2002 | 50 | 0.04 |
| 17 | Wales | 2000 | 16 | 0.04 |
| 18 | Belgium | 1996 | 40 | 0.03 |
| 19 | Portugal | 2014 | 30 | 0.03 |
| 20 | Ireland | 1999 | 16 | 0.03 |

## 4. Results and Discussions

### 4.1. Research Hotspots Analysis

#### 4.1.1. Keywords Co-Occurrence Analysis

Keywords co-occurrence analysis can reveal research hotspots and the evolution of hotspots in the field of agricultural and rural public governance and sustainable development. A high-frequency keywords co-occurrence network map on agricultural and rural

public governance and sustainable development is shown in Figure 7. Figure 7 shows that the map of the high-frequency keywords cooperative network consists of 752 nodes and 4460 lines, and the density of the co-occurrence network is 0.0158. The largest subnetwork member has 675 nodes, accounting for 89% of the 752 nodes. It is well demonstrated that the cooperative network relationship of keywords is quite close. At the same time, intuitively speaking, keywords such as sustainable development, rural development, management, policy, agriculture, and systems occur very frequently.

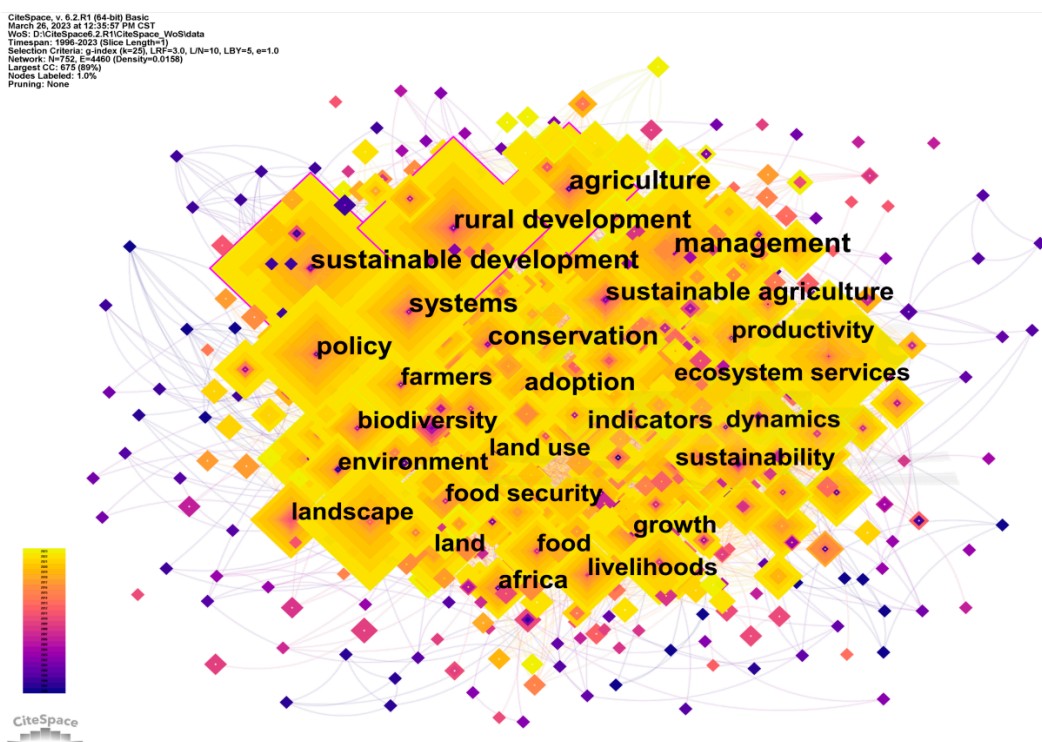

**Figure 7.** High-frequency keywords co-occurrence network map.

In view of the influence of nodes on the network structure, Table 4 collates the 20 keywords with the highest total number and concentration in agricultural and rural public governance and sustainable development. The keywords in the table are all high-frequency keywords with a high degree of centrality, and all are important turning points connecting different research fields, among which the keywords with the highest importance are as follows: sustainable development, rural development, agriculture, management, sustainable agriculture, policy, systems, conservation, and productivity. These also represent research hotspots in the field of agricultural and rural public governance and sustainable development.

### 4.1.2. Terms Cluster Knowledge Analysis

Cluster analysis can deeply explore the content and characteristics of the knowledge structure of agricultural and rural public governance and sustainable development research, highlight the key nodes and important connections, reflect the important positions that special points occupy in the knowledge network, and highlight the specific roles they play in the evolution of the knowledge structure. Figure 8 shows the cluster knowledge on agricultural and rural public governance and sustainable development. From the figure, the modularity Q for the cluster view is 0.3867, which is already more than 0.3; it indicates that the modular structure of the research network for agricultural and rural public governance and sustainable development is significant, and the clustering effect is better. The weighted mean silhouette S is 0.6741 for the cluster map, which is greater than 0.5 and close to 0.7; it proves that the clustering effect has a certain degree of reliability, and the clustering results

are convincing and reasonable. It can be said with certainty that ecosystem service (# 0), sustainable agriculture (# 1), land consolidation (# 2), agricultural intensification (# 3), and life cycle assessment (# 4), etc., represent the thematic areas that have received widespread attention and in-depth research in the field of agricultural research.

**Table 4.** The 20 keywords with the highest total number and centrality.

| Rank | Keywords | Year | Count | Centrality |
|------|----------|------|-------|------------|
| 1 | sustainable development | 1998 | 269 | 0.13 |
| 2 | rural development | 2000 | 243 | 0.11 |
| 3 | agriculture | 1996 | 127 | 0.11 |
| 4 | management | 1999 | 214 | 0.10 |
| 5 | sustainable agriculture | 2001 | 85 | 0.08 |
| 6 | policy | 1997 | 140 | 0.06 |
| 7 | systems | 2001 | 121 | 0.06 |
| 8 | conservation | 2003 | 99 | 0.06 |
| 9 | productivity | 1998 | 65 | 0.06 |
| 10 | farmers | 1999 | 75 | 0.05 |
| 11 | adoption | 2004 | 67 | 0.05 |
| 12 | indicators | 2002 | 32 | 0.05 |
| 13 | land use | 2000 | 121 | 0.04 |
| 14 | sustainability | 2009 | 69 | 0.04 |
| 15 | dynamics | 1996 | 50 | 0.04 |
| 16 | Africa | 1997 | 44 | 0.04 |
| 17 | technology | 2004 | 35 | 0.04 |
| 18 | environment | 1996 | 34 | 0.04 |
| 19 | deforestation | 2001 | 29 | 0.04 |
| 20 | agricultural policy | 2003 | 21 | 0.04 |

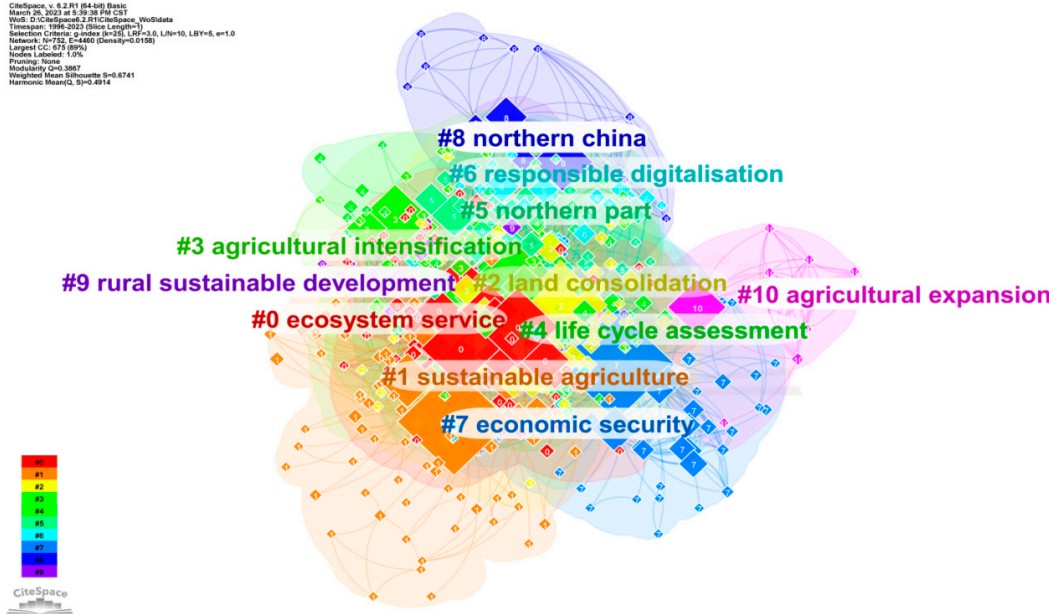

**Figure 8.** Terms cluster knowledge network map.

Table 5 provides a detailed ranking of the top term clusters and their label clustering information in agricultural and rural public governance and sustainable development. In general, the efficiency of each group of clustering groups is relatively high and the clustering effect is reasonable. Specifically, among each group of clusters, the cluster members extracted by the LLR algorithm are relatively significant, and these cluster units not only fully represent the characteristics, research mainstream, and their evolution in the research field of agricultural and rural public governance and sustainable development,

but also reflect the important position that special points occupy in the knowledge network of agricultural and rural public governance and sustainable development research.

**Table 5.** Top terms clusters and their label clustering information.

| Cluster ID | Size | Silhouette | Mean (Year) | Top Terms (Log-Likelihood Ratio, p-Level) |
|---|---|---|---|---|
| # 0 | 110 | 0.679 | 2009 | ecosystem service (1156.65, $1.0 \times 10^{-4}$); rural livelihood (800.04, $1.0 \times 10^{-4}$); and-burn agriculture (691.34, $1.0 \times 10^{-4}$); livelihood diversification (633.41, $1.0 \times 10^{-4}$); organic fertilizer investment (604.46, $1.0 \times 10^{-4}$) |
| # 1 | 108 | 0.718 | 2007 | sustainable agriculture (1035.08, $1.0 \times 10^{-4}$); social farming (663.33, $1.0 \times 10^{-4}$); agricultural policy (547.89, $1.0 \times 10^{-4}$); innovation network (489.38, $1.0 \times 10^{-4}$); organic food support (485.43, $1.0 \times 10^{-4}$) |
| # 2 | 104 | 0.506 | 2013 | land consolidation (1243.08, $1.0 \times 10^{-4}$); rural settlement (858.83, $1.0 \times 10^{-4}$); driving factor (812.88, $1.0 \times 10^{-4}$); urban sprawl (680.5, $1.0 \times 10^{-4}$); spatial distribution (631.92, $1.0 \times 10^{-4}$) |
| # 3 | 81 | 0.606 | 2013 | agricultural intensification (837.12, $1.0 \times 10^{-4}$); smart agriculture (768.57, $1.0 \times 10^{-4}$); democratic republic (699.77, $1.0 \times 10^{-4}$); sustainable intensification (544.14, $1.0 \times 10^{-4}$); household food security (480.01, $1.0 \times 10^{-4}$) |
| # 4 | 70 | 0.688 | 2012 | life cycle assessment (534.6, $1.0 \times 10^{-4}$); rethinking sustainability (418, $1.0 \times 10^{-4}$); viticulture realities (418, $1.0 \times 10^{-4}$); integrating economy landscape (418, $1.0 \times 10^{-4}$); environmental service (412.7, $1.0 \times 10^{-4}$) |
| # 5 | 67 | 0.633 | 2014 | northern part (658.88, $1.0 \times 10^{-4}$); forest resource utilization assessment (658.88, $1.0 \times 10^{-4}$); rural community (534.91, $1.0 \times 10^{-4}$); food insecurity (458.03, $1.0 \times 10^{-4}$); contextual change (427.4, $1.0 \times 10^{-4}$) |
| # 6 | 45 | 0.646 | 2015 | responsible digitalization (362.37, $1.0 \times 10^{-4}$); living lab (362.37, $1.0 \times 10^{-4}$); low-carbon agriculture (355.39, $1.0 \times 10^{-4}$); human advisory service (355.39, $1.0 \times 10^{-4}$); reducing nitrogen fertilizer use (355.39, $1.0 \times 10^{-4}$) |
| # 7 | 42 | 0.841 | 2011 | economic security (1406.04, $1.0 \times 10^{-4}$); information technologies (717.75, $1.0 \times 10^{-4}$); staff motivation (438.26, $1.0 \times 10^{-4}$); operational efficiency (432.41, $1.0 \times 10^{-4}$); education institution (427.72, $1.0 \times 10^{-4}$) |
| # 8 | 21 | 0.970 | 1998 | northern China (232.68, $1.0 \times 10^{-4}$); socioeconomic equity sustainability (227.21, $1.0 \times 10^{-4}$); carrying capacity (227.21, $1.0 \times 10^{-4}$); energy planning (217.63, $1.0 \times 10^{-4}$); Morogoro Tanzania (208.06, $1.0 \times 10^{-4}$) |
| # 9 | 16 | 0.924 | 2011 | rural sustainable development (196.51, $1.0 \times 10^{-4}$); sustainable requalification (191.67, $1.0 \times 10^{-4}$); traditional farm building (191.67, $1.0 \times 10^{-4}$); using analytic network process (191.67, $1.0 \times 10^{-4}$); southern Italy (191.67, $1.0 \times 10^{-4}$) |
| # 10 | 11 | 0.999 | 1998 | agricultural expansion (172.59, $1.0 \times 10^{-4}$); fertilizer use (170.67, $1.0 \times 10^{-4}$); increasing demand (161.25, $1.0 \times 10^{-4}$); agricultural support policies (149.96, $1.0 \times 10^{-4}$); farmers' perception (140.53, $1.0 \times 10^{-4}$) |

*4.2. Research Frontier Detection*

4.2.1. Reference Co-Citation Analysis

Reference co-citation analysis reflects the foundation of knowledge on research frontiers, which can reveal the knowledge structure of a certain research field and the evolution of its research frontiers, as well as the literature that plays a key role in the evolution process. Figure 9 depicts the cited reference network cluster map on agricultural and rural public governance and sustainable development. The information in the map structure clearly displays that the co-citation network consists of 965 nodes and 2357 lines, the density of the co-citation network is 0.0051, the modularity Q is 0.888, and the weighted mean silhouette S is 0.7872 for the co-citation cluster map, which fully reveals that the strength of the cooperation relationship of the co-citation network is very close, the association structure of the co-citation clustering is particularly remarkable, and the clustering effect is highly efficient and very convincing. Specifically speaking, the knowledge bases of agriculture, integrated systems, smallholder systems, Brazilian Amazon, rural sustainable development, and so on represent the research frontiers in agricultural and rural public

governance and sustainable development. Among them, the cited authors in the specific topic also condensed the academic community.

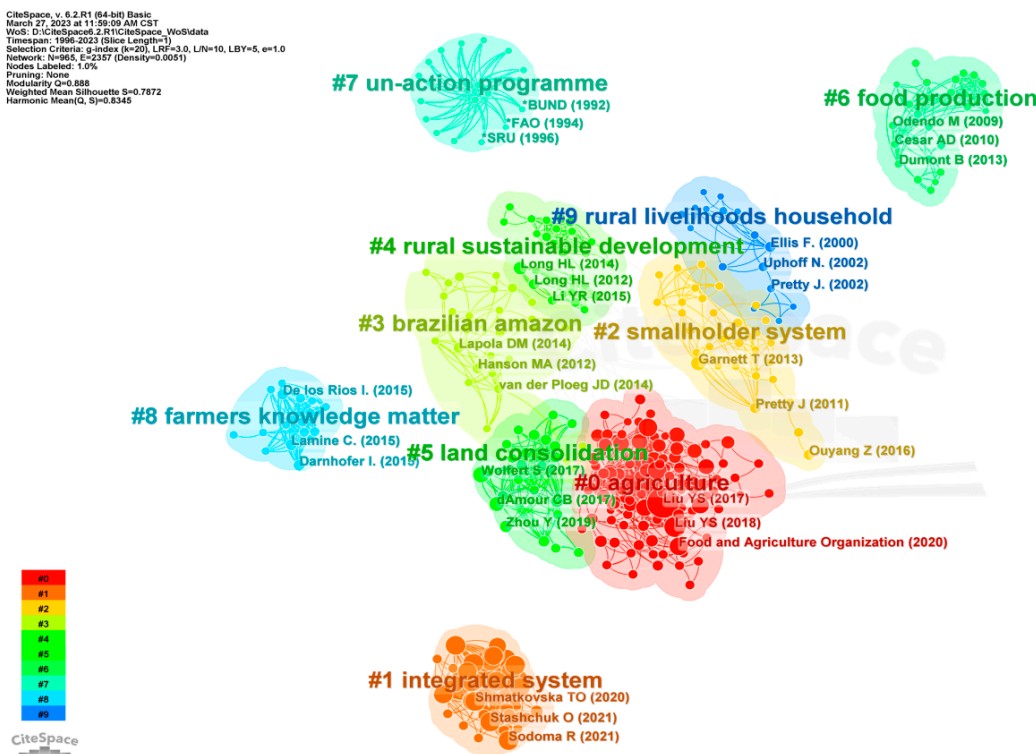

**Figure 9.** Cited reference network cluster map.

Table 6 lists the top term clusters and their label clustering information on reference co-citation in agricultural and rural public governance and sustainable development. The 10 major clusters, along with their size, silhouette, average year, and major labels, are clearly shown separately in Table 6. Taking cluster "# 0" as an example, there are 109 nodes in this cluster, of which the silhouette is 0.941, and the average publication year of citations is 2017. The representative cluster labels are as follows: case study (LSI algorithm), agriculture (LLR algorithm), and sensitivity analysis (MI algorithm).

### 4.2.2. The Strongest Sigma and Burstness Analysis

Sigma is a very important parameter that combines the indicators of betweenness centrality and burstness, which scientifically and reasonably reflect the influence of the knowledge node on the network structure, as well as the influence of the knowledge node on the time course. The larger the sigma of a certain paper, the more important the research area is, the higher the level of activity, and the more representative the emerging trend of research. Therefore, it is necessary to combine the references with the highest burstness and the highest sigma for comprehensive analysis. Table 7 combs out the top 11 references with the highest burstness, including title, journal, authors, publication time (year), strength, begin time of burstness (begin), end time of burstness (end) of references and range (1996–2023), and the red grid represents the duration of burstness of the references. Table 8 ranks the top eight references with the highest sigma. Tables 7 and 8 clearly tells us that knowledge foundations such as land consolidation, sustainable intensification in agriculture, land use, revitalize countryside, rural sustainability, and allocation and management of critical resources in rural, smallholder, and family farms, and so on have received special attention during the corresponding period since 2016, which to some extent also reflects the research frontier in the field of agricultural and rural public governance and sustainable development. In addition, it is worth mentioning that

in the past three years, smallholder farms, family farms, allocation, and management of critical resources in rural areas, the sustainability of agritourism activity, land use transitions and land management, and food security and nutrition have burst and become an emerging trend in the research field of agricultural and rural public governance and sustainable development.

**Table 6.** Top term clusters and their label clustering information for reference co-citation.

| Cluster ID | Size | Silhouette | Mean (Year) | Label (LSI) | Label (LLR) | Label (MI) |
|---|---|---|---|---|---|---|
| #0 | 109 | 0.941 | 2017 | case study | agriculture (67.58, $1.0 \times 10^{-4}$) | sensitivity analysis (6.05) |
| #1 | 46 | 1.000 | 2020 | management | integrated system (24.43, $1.0 \times 10^{-4}$) | case study (0.04) |
| #2 | 31 | 0.996 | 2013 | smallholder system | smallholder system (74.11, $1.0 \times 10^{-4}$) | central Malawi (0.04) |
| #3 | 30 | 0.917 | 2013 | Brazilian Amazon | Brazilian Amazon (78.88, $1.0 \times 10^{-4}$) | native vegetation protection law (0.05) |
| #4 | 27 | 0.962 | 2013 | rural sustainable development | rural sustainable development (82.23, $1.0 \times 10^{-4}$) | case study (0.03) |
| #5 | 27 | 0.960 | 2016 | land consolidation | land consolidation (148.15, $1.0 \times 10^{-4}$) | integrative approach (0.13) |
| #6 | 26 | 1.000 | 2010 | food production | food production (44.2, $1.0 \times 10^{-4}$) | case study (0.04) |
| #7 | 19 | 1.000 | 1994 | sustainable agriculture in agenda 21 | un-action program (15.6, $1.0 \times 10^{-4}$) | case study (0.05) |
| #8 | 18 | 1.000 | 2014 | resilient agriculture | farmers knowledge matter (27.42, $1.0 \times 10^{-4}$) | case study (0.04) |
| #9 | 16 | 1.000 | 2001 | development policy | rural livelihoods household (35.37, $1.0 \times 10^{-4}$) | case study (0.04) |

**Table 7.** Top 11 references with the highest burstness.

| Title | Journal | Authors | Year | Strength | Begin | End | Range (1996–2023) |
|---|---|---|---|---|---|---|---|
| Sustainable Intensification in Agriculture: Premises and Policies | SCIENCE | Garnett T | 2013 | 6.37 | 2016 | 2018 | |
| Land consolidation: An indispensable way of spatial restructuring in rural China | J GEOGR SCI | Long HL | 2014 | 3.98 | 2016 | 2019 | |
| Revitalize the world's countryside | NATURE | Liu YS | 2017 | 8.27 | 2019 | 2021 | |
| Introduction to land use and rural sustainability in China | LAND USE POLICY | Liu YS | 2018 | 6.47 | 2019 | 2021 | |
| Land consolidation for rural sustainability in China: Practical reflections and policy implications | LAND USE POLICY | Li YH | 2018 | 5.87 | 2019 | 2021 | |
| Targeted poverty alleviation and land policy innovation: Some practice and policy implications from China | LAND USE POLICY | Zhou Y | 2018 | 4.81 | 2019 | 2020 | |
| The Number, Size, and Distribution of Farms, Smallholder Farms, and Family Farms Worldwide | WORLD DEV | Lowder SK | 2016 | 7.99 | 2020 | 2021 | |
| The allocation and management of critical resources in rural China under restructuring: Problems and prospects | J RURAL STUD | Long HL | 2016 | 4.78 | 2020 | 2021 | |
| Sustainability of Agritourism Activity. Initiatives and Challenges in Romanian Mountain Rural Regions | SUSTAINABILITY | Adamov T | 2020 | 3.76 | 2020 | 2021 | |
| Land use transitions and land management: A mutual feedback perspective | LAND USE POLICY | Long HL | 2018 | 3.64 | 2020 | 2021 | |
| The State of Food Security and Nutrition in the World 2020 | FAO | FAO | 2020 | 3.58 | 2021 | 2023 | |

**Table 8.** Top 8 references with the highest sigma.

| Sigma | Burst | Centrality | Degree | Freq | Year | Label | Source | Cluster ID |
|---|---|---|---|---|---|---|---|---|
| 1.22 | 3.98 | 0.05 | 17 | 8 | 2014 | Long HL (2014) | J GEOGR SCI | # 4 |
| 1.20 | 6.37 | 0.03 | 15 | 11 | 2013 | Garnett T (2013) | SCIENCE | # 2 |
| 1.08 | 5.87 | 0.01 | 36 | 13 | 2018 | Li YH (2018) | LAND USE POLICY | # 0 |
| 1.06 | 8.27 | 0.01 | 28 | 52 | 2017 | Liu YS (2017) | NATURE | # 0 |
| 1.05 | 6.47 | 0.01 | 27 | 22 | 2018 | Liu YS (2018) | LAND USE POLICY | # 0 |
| 1.04 | 4.78 | 0.01 | 27 | 9 | 2016 | Long HL (2016) | J RURAL STUD | # 0 |
| 1.01 | 3.64 | 0.00 | 25 | 16 | 2018 | Long HL (2018) | LAND USE POLICY | # 0 |
| 1.01 | 7.99 | 0.00 | 8 | 15 | 2016 | Lowder SK (2016) | WORLD DEV | # 0 |

### 4.3. Research Evolution Path

The timeline view delineates the relationships between the clusters and the historical span of the literature in a certain cluster, reveals the interconnections and mutual influences between the clusters, and reflects the time span of the research basis for a certain research topic. Figure 10 draws the timeline view map of the research cluster. From the figure, the information in the timeline view map structure clearly displays that the network consists of 621 nodes and 3714 lines, the density of the co-citation network is 0.0193, the largest subnetwork member has 563 nodes, accounting for 90% of the 621 nodes, the modularity Q is 0.3765, and the weighted mean silhouette S is 0.727 for the timeline view map, which means the results are convincing and the results are reliable.

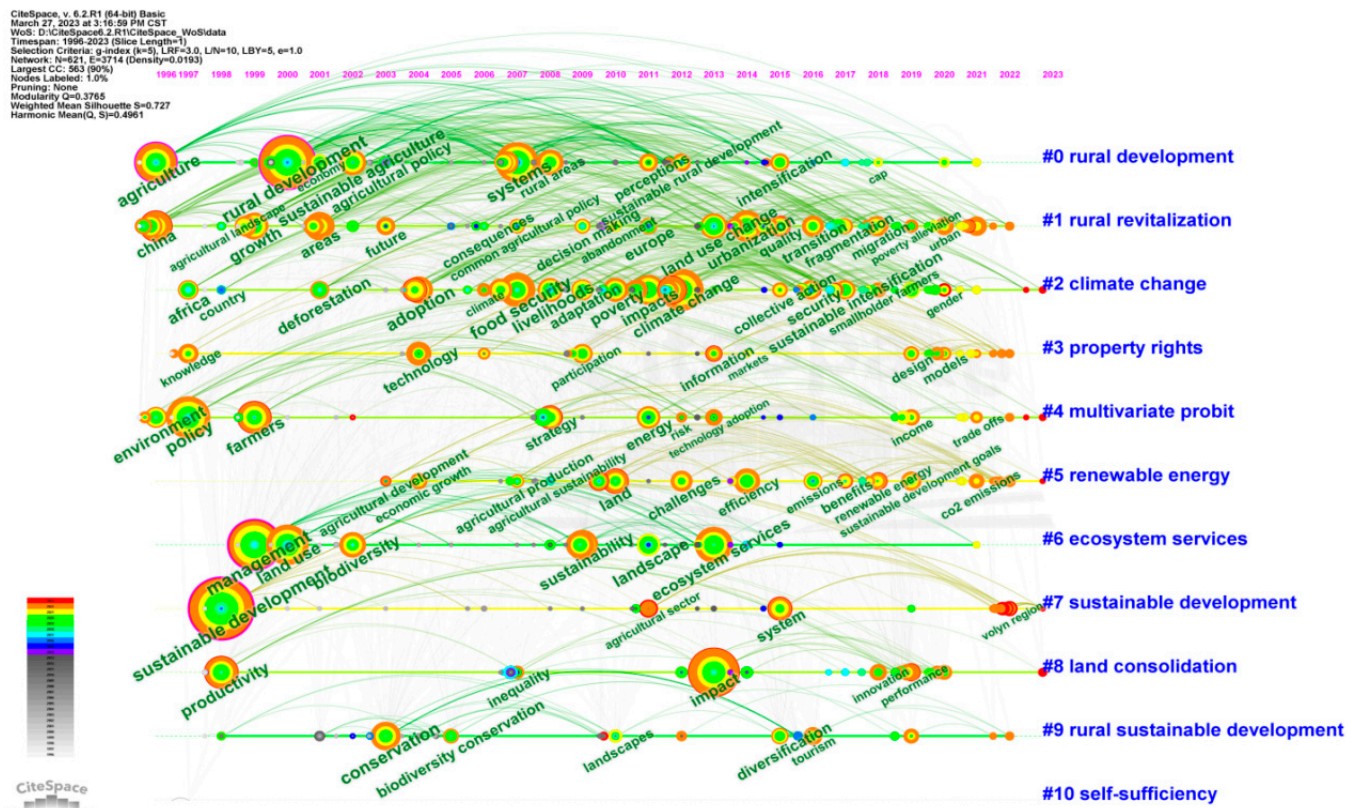

**Figure 10.** Timeline view map of research cluster.

Firstly, in 1996, the clusters of "# 0 rural development", "# 1 rural revitalization", and "#4 multivariate probit" began to appear, that is, the above clusters already have their first references. In 1997, the clusters of "# 2 climate change" and "# 3 property rights" began to appear. In 1998, the clusters of "# 7 sustainable development", "# 8 land consolidation", and "# 9 rural sustainable development" began to appear. In 1999, the clusters of "# 6

ecosystem services" began to appear. In 2003, the clusters of "# 5 renewable energy" began to appear.

Secondly, in 2000, the research results for cluster "# 0 rural development" began to increase. In 2013, the research results for cluster "# 1 rural revitalization" began to grow. In 2012, the research results for cluster "# 2 climate change" began to rise. In 2019, the research results for cluster "# 5 renewable energy" began to rise. In 1998, the research results for cluster "# 7 sustainable development" began to increase.

Thirdly, in cluster "# 0 rural development", landmark literature on topics such as "agriculture" and "rural development" emerged in 1996 and 2000, respectively, which occupy an important position in the whole cluster and influenced the trend of the whole cluster. Among cluster "# 6 ecosystem services", landmark literature on "management" appeared in 1999, which plays an important role in the whole cluster. In cluster "# 7 sustainable development", the landmark literature on "sustainable development" appeared in 1998, which plays a significant role in the overall cluster and influences the trend of the overall cluster.

## 5. Conclusions

This paper was based on the research topic of agricultural and rural public governance and sustainable development. With the help of CiteSpace metrological analysis software, a scientific metrological and knowledge map analysis was conducted on spatial–temporal evolution, collaboration networks, research hotspots, cluster labels, frontier detection, and the evolution path of 2350 pieces of data. The main results of this paper are summarized as follows.

(1) Beginning at the end of the 20th century, scholars began to focus on the field of agricultural and rural public governance and sustainable development and conduct research. Since 2015, there has been an extremely rapid increase in the number of publications in the literature. The core research force is mainly distributed in the Chinese Academy of Sciences, Wageningen University & Research, and CGIAR. The 10 countries/regions with the highest node centrality are England, USA, Germany, Australia, Italy, Canada, Slovakia, Netherlands, People's R. China, and India. The collaboration network among authors, between institutions, and between countries/regions each had a certain degree of connection in different periods, respectively.

(2) The structure of the knowledge map analysis is significant, and the results are highly reliable. The count for many research hotspots, such as sustainable development, rural development, agriculture, and others, have influenced the development of the entire research process and have evolved into larger topic cluster groups such as ecosystem services, sustainable agriculture, land consolidation, and agricultural intensification. These have evolved into research frontiers and knowledge foundations in which the following structures are significant and play key roles: agriculture, integrated systems, smallholder systems, Brazilian Amazon, rural sustainable development, and land consolidation. Smallholder farms, family farms, critical resources in rural areas, the sustainability of agritourism activity, land use transitions, and food security and nutrition have seen an emerging trend in the last three years.

(3) Agriculture (1996), sustainable development (1998), management (1999), and rural development (2000) occupy an important position in their corresponding clusters and have influenced the trend of the whole cluster in each corresponding period, respectively. In the last three years, research clusters that have received continuous attention include rural revitalization, climate change, property rights, multivariate probit, renewable energy, and sustainable development. Moving toward sustainable development is a common and unchanging issue for humanity.

Based on the above findings, although agricultural and rural public governance and sustainable development is increasingly receiving high attention in academic research and practice environments, there are still many challenges in the research field of agricultural and rural public governance and sustainable development, for which further development

deeply is urgently required. Firstly, this would be through deepening academic exchanges and strengthening academic cooperation. In this paper, the map of the author cooperative network consists of 805 nodes and 502 lines, and the density of the co-occurrence network is 0.0016, which shows that the cooperation intensity needs to be strengthened. Therefore, scholars from different countries or regions need to deepen their cooperation network constantly, making academic contributions to the research field of agricultural and rural public governance and sustainable development to achieve the goal of global sustainable development, and to jointly build and consolidate the academic community with close cooperation. Secondly, promoting in-depth research on important issues in the field of agricultural and rural public governance and sustainable development is necessary. In this paper, the keywords co-occurrence analysis results show that the three keywords with the highest centrality are sustainable development, rural development, and agriculture. The terms cluster knowledge analysis results show that the three term clusters with the highest label value are ecosystem service, sustainable agriculture, and land consolidation. The reference co-citation analysis shows that the three term clusters with the highest label value (LLR) are agriculture, integrated systems, and smallholder systems. The strongest sigma and burstness analysis results show that smallholder farms, family farms, rural critical resources, agritourism activity, land use and management, and food security and nutrition have burst and become an emerging trend in the past three years. At present, there is a certain consensus in the academic circle around the research theme of agricultural and rural public governance and sustainable development, but it is still necessary to focus on the UN 2030 SDGs, combining the countries' and regions' development needs and realities, and carrying out further in-depth discussions on research fields such as sustainable agriculture, smallholder systems, food production and security, land consolidation, rural sustainable development, and agricultural green total factor productivity [44], to then put forward countermeasures which conform to the mainstream norms and can be used for reference. Thirdly, it is necessary to strengthen public governance in agricultural and rural areas. In this paper, the evolution path analysis results show 10 clusters and their keywords, which actually reflect the public governance issues that have important impacts and are of great concern in the process of agricultural and rural sustainable development. Therefore, the governance ability and governance level of sustainable development in global agricultural and rural areas urgently need to be improved. It is important to further clarify and refine the topics that need to be studied and the problems that need to be solved. More scientific demonstration and more feasible measures should be adopted to jointly deal with and enhance awareness of the current problems and practical challenges and further promote practical development by fostering academic consensus and expanding and innovating the governance models from the comprehensive dimensions of economy, politics, society, culture, and ecological environment so as to achieve good agricultural and rural governance.

This paper has some limitations and future research suggestions, which are as follows: Firstly, this paper presents research on agricultural and rural public governance and sustainable development with 2350 pieces of data from the core collection database of the Web of Science, which has high authority and persuasiveness. However, the core database is basically updated every day. In the future, study samples should require data sources to be updated to provide analysis results that keep pace. Secondly, the scientific metrological analysis of agricultural and rural public governance and sustainable development is mainly based on data from the published literature, and the findings are considered to be of high confidence. However, this study lacks a combined analysis of practical cases. In the future, we will consider innovation and fusion research with practical cases.

**Author Contributions:** T.H. and Q.H. conceived and designed the research question. T.H. constructed the models and wrote the paper. Q.H. reviewed and edited the manuscript. All authors have read and agreed to the published version of the manuscript.

**Funding:** This research was funded by the National Social Science Foundation of China (grant no.: 22BJY037), and the Project of Guizhou Provincial Department of Education (grant no.: 2022ZX010).

**Institutional Review Board Statement:** Not applicable.

**Informed Consent Statement:** Not applicable.

**Data Availability Statement:** The data that support the finding of this study are available from the corresponding author upon reasonable request.

**Acknowledgments:** The authors are grateful to the anonymous referees who provided valuable comments and suggestions to significantly improve the quality of the paper. We gratefully acknowledge the National Social Science Foundation of China (grant no.: 22BJY037), the Project of Guizhou Provincial Department of Education (grant no.: 2022ZX010).

**Conflicts of Interest:** The authors declare no conflict of interest. There is no professional or other personal interest of any nature or kind in any product, service, and/or company that could be construed as influencing the position presented in, or the review of, the manuscript entitled.

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
