# Peer review of "Research on Agricultural and Rural Public Governance and Sustainable Development: Evidence from 2350 Data"

_sustainability, doi:10.3390/su15107876_

Round 1

Reviewer 1 Report

the paper is very interesting and is based on an evaluation of the production of scientific literature on the themes of agricultural and rural public governance and sustainable development, using indicators related to network analysis. the link between the field of research and the method of analysis is particularly innovative and this represents a strong point of work. Nevertheless, I think it is essential, in order to make the paper more accessible:

1) a greater descriptive clarity of the methodology and indicators identified, in particular with regards to the subject of the paper; in the paragraphs relating to the discussion of the results the use of the indicator is taken for granted but is almost never clear;

2) only a mention is made of the growing dynamics of work over the period within paragraph 3.1, but for the remaining indicators the evolutionary aspect is neglected; could be a choice of authors but it might be interesting to consider in the discussion and conclusions also the evolutionary aspect;

3) the reading of the text does not flow fluently and I think it needs improvement (as I said I am not an expert so my opinion is just advice)

4) in the text is repeated many times "agricultural and rural public governance and sustainable development" and I think it is necessary to find synonyms that can avoid repetition.

5) some tables, especially Table 7, are difficult to read and you should work on either editing or a more precise description in the text.

Author Response

Dear Editors and Reviewers:

Thank you for your letter and for the reviewer’s comments concerning our manuscript entitled “Research on Agricultural and Rural Public Governance and Sustainable Development: Evidence from 2350 Data” (ID: sustainability-2378110). Those comments are all valuable and very helpful for revising and improving our paper, as well as the important guiding significance to our researches. We have studied comments carefully and have made correction which we hope meet with approval. Due to the heavy workload of revision, we have tried our best to revise our manuscript according to the comments. Attached please find the revised version, which we would like to submit for your kind consideration.

The main corrections in the paper and the responds to the reviewer’s comments are as flowing:

Reviewer #1:

  1. Response to comment: ( a greater descriptive clarity of the methodology and indicators identified, in particular with regards to the subject of the paper; in the paragraphs relating to the discussion of the results the use of the indicator is taken for granted but is almost never clear;)

Response:

Thanks for the Reviewer’s comments. However, in fact, we have had the corresponding introduction in the last part of the introduction, concept model and research method. Considering the Reviewer’s comments, we have optimized the relevant expression content, and see red font content on page 3-4 for more details.

In light of this, the purpose of this paper is to focus on taking agricultural and rural public governance and sustainable development as the research theme, which based on the WoS core collection database and CiteSpace metrological analysis software, provides an in-depth examination of the spatial-temporal evolution, cutting-edge map, and logical evolution of research on public governance and sustainable development in agriculture and rural areas. Firstly, it can contribute to reveal and reflect the status and progress, the topics and hot spots, as well as the mainstream and fronts, and the trends and vistas of its research field from a systematic, comprehensive, and entirety perspective for research on public governance and sustainable development in agriculture and rural areas. Secondly, it can contribute to fully grasp the logic context of the development history and the frontier trend of hot knowledge for research on public governance and sustainable development in agriculture and rural areas. Thirdly, it can contribute to provide academic perspectives and research foundations for other scholars or decision-makers to engage in related research of agricultural and rural public governance and sustainable development.

This paper is referred to drawing on the concept model of CiteSpace pioneered by Prof. Chaomei Chen [41,42], this model creatively integrates the methods of citation analysis (diachronic) and co-citation analysis (structural), creates the mapping from knowledge base to research frontier, and highlights the discipline basis of citation network map and the technical basis of information space map [42-44]. With the help of version V.6.2.R1 of the CiteSpace that specialized scientific metrological analysis software with relatively complete functions, this paper is expected to analyze the basic situation, research hotspots, frontier detection, and evolution path of agricultural and rural public governance and sustainable development, in order to find out the knowledge of potential associations contained in the research on agricultural rural public governance and sustainable development, and characterize a series of visual knowledge maps, so as to reveal and reflect the multivariate, time-sharing, and dynamic spatial information, as well as its network relationships and mutual influences, and then to interrogate the temporal and spatial variations, frontier probing and evolution paths in the filed of agricultural rural public governance and sustainable development. The conceptual model of CiteSpace is shown in Figure 1.

In this paper, according to the research goal and conceptual model, an application model of CiteSpace for research on public governance and sustainable development in agriculture and rural areas was created. The application model of CiteSpace on agricultural and rural public governance and sustainable development is depicted in Figure 2. Furthermore, this paper attempts to answer the following questions:

(1) When did the field of agricultural and rural public governance and sustainable development start to be studied? Where is the research stronger? What well-known scholars are paying attention to? What is the cooperation network in this field?

(2) What research hotspots have emerged in the field of agricultural and rural public governance and sustainable development? What research topic clusters have been cohered? What research frontiers are presented?

(3) How has the knowledge base and research paradigm in the field of agricultural and rural public governance and sustainable development evolved? In which years did a cluster appear? In which years did the research results of a certain cluster begin to increase or decrease? In which years did iconic literature appear and affect the overall trend of clustering?

Among them, the important calculation formulas involved are introduced as follows.

  1. Response to comment: ( only a mention is made of the growing dynamics of work over the period within paragraph 3.1, but for the remaining indicators the evolutionary aspect is neglected; could be a choice of authors but it might be interesting to consider in the discussion and conclusions also the evolutionary aspect;)

Response:

Considering the Reviewer’s suggestion, we have revised and improved this part of “3.1. Trends in the Number of Published Papers”, and see red font content on page 7 for more details.

Figure 3 depicts the trend of published papers on agricultural and rural public governance and sustainable development, which helps to intuitively grasp the publication dynamics of research on public governance and sustainable development in agriculture and rural areas. From the overall characteristics, since 1996, the core literature of agricultural and rural public governance and sustainable development research has appeared, the fluctuation in the number of publications in the literature on agricultural and rural public governance and sustainable development is very flat between 1996 and 2015, with no more than 70 publications in the highest annual volume. However, it is surprising that from 2016 to 2022, the number of literature publications showed a rapid linear increase. During this period, the lowest annual publication volume was 81 in 2016 and reached the highest annual publication volume of 420 in 2022. This means that after 2015, the number of publications of research field in the agricultural and rural public governance and sustainable development showes a steep slope growth, which has a great relationship with the UN summit on Sustainable Development in 2015 adopted 17 sustainable development goals with the aim of turning to the path of sustainable development, and has also caused in-depth research and extensive discussion on this theme of the academic community.

  1. Response to comment: ( the reading of the text does not flow fluently and I think it needs improvement (as I said I am not an expert so my opinion is just advice))

Response:

Thanks for the Reviewer’s comments. Considering the Reviewer’s suggestion, we have optimized the English language expression of this manuscript, and see red font content for more details. 

  1. Response to comment: ( in the text is repeated many times "agricultural and rural public governance and sustainable development" and I think it is necessary to find synonyms that can avoid repetition. )

Response:

We are grateful to your comments for the “agricultural and rural public governance and sustainable development”. Because “agricultural and rural public governance and sustainable development” is the research theme of this paper, and data sources are accessed based on this theme, if the theme is replaced, it is likely that the ambiguity or diversity of the theme will lead to deviations between the data results and analysis. So, we sincerely hope to gain your understanding.

  1. Response to comment: ( some tables, especially Table 7, are difficult to read and you should work on either editing or a more precise description in the text.)

Response:

Thanks for the Reviewer’s comments. Considering the Reviewer’s suggestion, we have revised and improved this part of “4.2.2. The Strongest Sigma and Burstness Analysis”, and see red font content on page 17 for more details.

Sigma is a very important parameter that combines the indicators of betweenness centrality and burstness, which scientifically and reasonably reflect the influence of the knowledge node in the network structure, as well as the influence of the knowledge node in the time course. When the larger the sigma of a certain literature, the more important the research area is, higher the level of activity, and the more representative the emerging trend of research. Therefore, it is necessary to combine the highest burstness and the highest sigma  of references for comprehensive analysis. Table 7 combs out the top 11 references with the highest burstness, including title, journal, authors, publication time (year), strength, begin time of burstness (begin), end time of burstness (end) of references and range (1996 - 2023), and the red grid represents the duration of burstness of references. Table 8 ranks the top 8 references with the highest sigma. Table 7 and Table 8 clearly tells us that, the following knowledge foundations such as land consolidation, sustainable intensification in agriculture, land use, revitalize countryside, rural sustainability, allocation and management of critical resources in rural, smallholder farms, and family farms, and so on, have received special attention during the corresponding period since 2016, which to some extent also means the research frontier in the field of agricultural and rural public governance and sustainable development. In addition, it is worth mentioning that in the recent three years, smallholder farms, family farms, allocation and management of critical resources in rural, sustainability of agritourism activity, land use transitions and land management, food security and nutrition have burst and become an emerging trend in research field of agricultural and rural public governance and sustainable development.

Thanks very much for your kind work and consideration on publication of our paper. On behalf of my co-authors, we would like to express our great appreciation to editor and reviewers.

Thank you and best regards.

Sincerely yours,

Name: Tingting Huang

E-mail: Sherohtt@foxmail.com

Reviewer 2 Report

This paper systematically reviews the existing research results in the field of Agricultural and Rural Public Governance and Sustainable Development, which would help subsequent researchers understand the research progress and provide reference for future research. However, the research depth of this paper needs to be further strengthened:

On the one hand, for example 3.4. Country/Region Cooperation Networks. Compared with the cooperation network, more important: which countries or regions the existing research has been carried out in; and taking China as an example, scholars from which countries have carried out relevant research in China.

On the another hand, in 4 Results and Discussions: This part of the content is just a brief description of the results, there is no in-depth discussion, and it needs to be strengthened.

Minor editing of English language required

Author Response

Dear Editors and Reviewers:

Thank you for your letter and for the reviewer’s comments concerning our manuscript entitled “Research on Agricultural and Rural Public Governance and Sustainable Development: Evidence from 2350 Data” (ID: sustainability-2378110). Those comments are all valuable and very helpful for revising and improving our paper, as well as the important guiding significance to our researches. We have studied comments carefully and have made correction which we hope meet with approval. Due to the heavy workload of revision, we have tried our best to revise our manuscript according to the comments. Attached please find the revised version, which we would like to submit for your kind consideration.

The main corrections in the paper and the responds to the reviewer’s comments are as flowing:

Reviewer #2:

  1. Response to comment: ( On the one hand, for example 3.4. Country/Region Cooperation Networks. Compared with the cooperation network, more important: which countries or regions the existing research has been carried out in; and taking China as an example, scholars from which countries have carried out relevant research in China.)

Response:

Thanks for the Reviewer’s comments. Considering the Reviewer’s suggestion, we have optimized the part of “3.4. Country/Region Cooperation Networks”, and see red font content on page 10-11 for more details.

Network analysis of country/region cooperation can outline the distribution of cooperation and the strength of cooperation relationships among different countries in the field of agricultural and rural public governance and sustainable development. Figure 6 plainly portrays the country/region cooperation network map on agricultural and rural public governance and sustainable development. From the information of map structure, the map of the country/region cooperative network consists of 136 nodes and 1175 lines, and the density of the co-occurrence network is 0.128. It reflects that the cooperative relationship between countries has a relatively high strength. In particular, since 1996, the research field of agricultural and rural public governance and sustainable development has occurred at a relatively high frequency in the following countries/regions, namely PEOPLES R CHINA, USA, ENGLAND, ITALY, GERMANY, NETHERLANDS, AUSTRALIA, INDIA, SPAIN, and FRANCE.

Table 3 lists the 20 countries or regions with the highest total number and centrality on agricultural and rural public governance and sustainable development. The ten countries/regions with the highest node centrality are ENGLAND, USA, GERMANY, AUSTRALIA, ITALY, CANADA, SLOVAKIA, NETHERLANDS, PEOPLES R CHINA and INDIA, respectively, which also illustrates that the above countries/regions have higher impacts in agricultural and rural public governance and sustainable development research.

  1. Response to comment: ( On the another hand, in 4 Results and Discussions: This part of the content is just a brief description of the results, there is no in-depth discussion, and it needs to be strengthened.)

Response:

Thanks for the Reviewer’s comments. Considering the Reviewer’s suggestion, we have optimized the relevant expression content of “4. Results and Discussions”, and see red font content for more details.

Thanks very much for your kind work and consideration on publication of our paper. On behalf of my co-authors, we would like to express our great appreciation to editor and reviewers.

Thank you and best regards.

Sincerely yours,

Name: Tingting Huang

E-mail: Sherohtt@foxmail.com

Reviewer 3 Report

Thank you for an important manuscript in the field of improving the efficiency of public administration in agriculture based on the development of a concept in accordance with the needs of the regions, an integrated approach and sustainable development goals. It is necessary to note an interesting approach based on the software for metrological analysis CiteSpace, which was used in the work. In order to make it more valuable to readers, I suggest doing a deep correspondence in several places.

The document does not contain a clear goal of the study, tasks. The authors do not provide any research questions (or hypotheses) for testing. I propose to revise the methodological part of the manuscript and supplement it with some research questions (hypotheses), the answers to which are confirmed by the results presented in the article.

The introduction provides a fairly broad literature review, but special attention should be paid to the analysis of the legal framework of state regulation both in China and in other countries.

Section 3.3. it is necessary to more clearly justify the criteria for constructing a sample network of research institutions. It would also be interesting to study the sectoral profile of research and build the corresponding cluster groups.

In general, there is a feeling of a gap between the results obtained and their further application. Thus, the conclusions recommend strengthening ties in academic cooperation. However, it is not specified with the help of what tools and mechanisms it is possible to implement this. It is also not clear on the basis of what criteria the gradation of the needs of countries and regions for sustainable development is possible, how these needs can be investigated and requests for the development of appropriate programs at the regional and municipal levels of government can be obtained, there is not enough justification for logical schemes and mechanisms for improving the efficiency of public administration based on the obtained significant the results of the study.

In conclusion, it is necessary to highlight the scientific novelty and practical significance of the study.

I hope my suggestions will improve the text.

Some phrases are very long and difficult to understand. It is recommended to improve the translation and present it in clearer simple sentences.

Author Response

Dear Editors and Reviewers:

Thank you for your letter and for the reviewer’s comments concerning our manuscript entitled “Research on Agricultural and Rural Public Governance and Sustainable Development: Evidence from 2350 Data” (ID: sustainability-2378110). Those comments are all valuable and very helpful for revising and improving our paper, as well as the important guiding significance to our researches. We have studied comments carefully and have made correction which we hope meet with approval. Due to the heavy workload of revision, we have tried our best to revise our manuscript according to the comments. Attached please find the revised version, which we would like to submit for your kind consideration.

The main corrections in the paper and the responds to the reviewer’s comments are as flowing:

Reviewer #3:

  1. Response to comment: ( The document does not contain a clear goal of the study, tasks. The authors do not provide any research questions (or hypotheses) for testing. I propose to revise the methodological part of the manuscript and supplement it with some research questions (hypotheses), the answers to which are confirmed by the results presented in the article. )

Response:

Thanks for the Reviewer’s comments. According to the Reviewer’s comments, we have improved the goal and tasks of the study, and supplemented the research questions in this paper, and see red font content on page 3-4 for more details.

In light of this, the purpose of this paper is to focus on taking agricultural and rural public governance and sustainable development as the research theme, which based on the WoS core collection database and CiteSpace metrological analysis software, provides an in-depth examination of the spatial-temporal evolution, cutting-edge map, and logical evolution of research on public governance and sustainable development in agriculture and rural areas. Firstly, it can contribute to reveal and reflect the status and progress, the topics and hot spots, as well as the mainstream and fronts, and the trends and vistas of its research field from a systematic, comprehensive, and entirety perspective for research on public governance and sustainable development in agriculture and rural areas. Secondly, it can contribute to fully grasp the logic context of the development history and the frontier trend of hot knowledge for research on public governance and sustainable development in agriculture and rural areas. Thirdly, it can contribute to provide academic perspectives and research foundations for other scholars or decision-makers to engage in related research of agricultural and rural public governance and sustainable development.

This paper is referred to drawing on the concept model of CiteSpace pioneered by Prof. Chaomei Chen [41,42], this model creatively integrates the methods of citation analysis (diachronic) and co-citation analysis (structural), creates the mapping from knowledge base to research frontier, and highlights the discipline basis of citation network map and the technical basis of information space map [42-44]. With the help of version V.6.2.R1 of the CiteSpace that specialized scientific metrological analysis software with relatively complete functions, this paper is expected to analyze the basic situation, research hotspots, frontier detection, and evolution path of agricultural and rural public governance and sustainable development, in order to find out the knowledge of potential associations contained in the research on agricultural rural public governance and sustainable development, and characterize a series of visual knowledge maps, so as to reveal and reflect the multivariate, time-sharing, and dynamic spatial information, as well as its network relationships and mutual influences, and then to interrogate the temporal and spatial variations, frontier probing and evolution paths in the filed of agricultural rural public governance and sustainable development. The conceptual model of CiteSpace is shown in Figure 1.

In this paper, according to the research goal and conceptual model, an application model of CiteSpace for research on public governance and sustainable development in agriculture and rural areas was created. The application model of CiteSpace on agricultural and rural public governance and sustainable development is depicted in Figure 2. Furthermore, this paper attempts to answer the following questions:

(1) When did the field of agricultural and rural public governance and sustainable development start to be studied? Where is the research stronger? What well-known scholars are paying attention to? What is the cooperation network in this field?

(2) What research hotspots have emerged in the field of agricultural and rural public governance and sustainable development? What research topic clusters have been cohered? What research frontiers are presented?

(3) How has the knowledge base and research paradigm in the field of agricultural and rural public governance and sustainable development evolved? In which years did a cluster appear? In which years did the research results of a certain cluster begin to increase or decrease? In which years did iconic literature appear and affect the overall trend of clustering?

  1. Response to comment: ( The introduction provides a fairly broad literature review, but special attention should be paid to the analysis of the legal framework of state regulation both in China and in other countries.)

Response:

Thanks for the Reviewer’s comments. But this article does not seem to involve the issues of legal framework of state regulation. We sincerely hope that the reviewer can provide clear guidance and correction.

  1. Response to comment: ( Section 3.3. it is necessary to more clearly justify the criteria for constructing a sample network of research institutions. It would also be interesting to study the sectoral profile of research and build the corresponding cluster groups.)

Response:

We are grateful to the Reviewer’s comments. In this paper, according to the conceptual model of CiteSpace, the overall analysis is based on one sample with 2350 data, and the network analysis of research institutions is also based on these 2350 data.

  1. Response to comment: ( In general, there is a feeling of a gap between the results obtained and their further application. Thus, the conclusions recommend strengthening ties in academic cooperation. However, it is not specified with the help of what tools and mechanisms it is possible to implement this. It is also not clear on the basis of what criteria the gradation of the needs of countries and regions for sustainable development is possible, how these needs can be investigated and requests for the development of appropriate programs at the regional and municipal levels of government can be obtained, there is not enough justification for logical schemes and mechanisms for improving the efficiency of public administration based on the obtained significant the results of the study.)

Response:

We are grateful to the Reviewer’s comments. Considering the Reviewer’s suggestion, we have optimized the recommendation of the study, and see red font content on page 20 for more details.

Based on the above findings, although agricultural and rural public governance and sustainable development is increasingly receiving high attention in academic research and practice environments, there are still many challenges in the research field of agricultural and rural public governance and sustainable development, which further development deeply is urgently required. Firstly, scholars from different countries or regions need to deepen their cooperation network constantly, make academic contributions in the research field of agricultural and rural public governance and sustainable development to achieve the goal of global sustainable development, and jointly build and consolidate the academic community with close cooperation. Secondly, at present, there is a certain consensus in the academic circle around the theme research of agricultural and rural public governance and sustainable development, but it is still necessary to focus on the UN 2030 SDGs, combine the countries and regional development needs and reality, carry out further in-depth discussion in the following research fields, such as sustainable agriculture, smallholder system, food production, rural sustainable development, agricultural green total factor productivity, and anti-poverty, and then put forward countermeasures which conform to the mainstream norms and can be used for reference. Thirdly, governance ability and governance level of sustainable development of global agriculture and rural areas urgently need to be improved. Further clarify and refine the topics that need to be studied and the problems that need to be solved, more scientific demonstration and more feasible measures should be adopted to jointly deal with and solve the current awareness problems and practical challenges, and further promote practical development by cohering academic consensus, expand and innovate the governance model from the comprehensive dimensions of economy, politics, society, culture and ecological environment so as to achieve good governance of agricultural and rural.

  1. Response to comment: ( In conclusion, it is necessary to highlight the scientific novelty and practical significance of the study.)

Response:

We are grateful to the Reviewer’s comments. Considering the Reviewer’s suggestion, we have highlighted the scientific novelty and practical significance of the study, and see red font content on page 3 for more details.

In light of this, the purpose of this paper is to focus on taking agricultural and rural public governance and sustainable development as the research theme, which based on the WoS core collection database and CiteSpace metrological analysis software, provides an in-depth examination of the spatial-temporal evolution, cutting-edge map, and logical evolution of research on public governance and sustainable development in agriculture and rural areas. Firstly, it can contribute to reveal and reflect the status and progress, the topics and hot spots, as well as the mainstream and fronts, and the trends and vistas of its research field from a systematic, comprehensive, and entirety perspective for research on public governance and sustainable development in agriculture and rural areas. Secondly, it can contribute to fully grasp the logic context of the development history and the frontier trend of hot knowledge for research on public governance and sustainable development in agriculture and rural areas. Thirdly, it can contribute to provide academic perspectives and research foundations for other scholars or decision-makers to engage in related research of agricultural and rural public governance and sustainable development.

  1. Response to comment: ( Some phrases are very long and difficult to understand. It is recommended to improve the translation and present it in clearer simple sentences.)

Response:

Thanks for the Reviewer’s comments. Considering the Reviewer’s suggestion, we have optimized the English language expression of this manuscript, and see red font content for more details.

Thanks very much for your kind work and consideration on publication of our paper. On behalf of my co-authors, we would like to express our great appreciation to editor and reviewers.

Thank you and best regards.

Sincerely yours,

Name: Tingting Huang

E-mail: Sherohtt@foxmail.com

Reviewer 4 Report

In this paper, 2350 pieces of literature had been reviewed on Agricultural and Rural Public Governance and Sustainable Development.

The subject of the paper is quite interesting. However, exciting and satisfying results were not presented in the paper.

Such as, The hypothesis of the paper has not been fully revealed.

Despite 2350 literature review, 39 references (39th and 40th references are the same) were used. These references generally belong to the regions of China and Africa.

However, the authors stated that there has been an extremely rapid increase in the number of publications in the literature since 2015, and the Core research power is mainly distributed in the Chinese Academy of Sciences, Wageningen University and Research and CGIAR. They said that the countries with superior influence status are mainly UK, USA, GERMANY, AUSTRALIA and ITALY.

As a result, the study could not go beyond a situation determination. Authors should explain exactly what they want to reveal with this study. Also, what should be done about the subject of the paper, what is the problem or what is missing?

The results obtained in the paper should be discussed more concretely.

Thank you,

Sincerely.

Author Response

Dear Editors and Reviewers:

Thank you for your letter and for the reviewer’s comments concerning our manuscript entitled “Research on Agricultural and Rural Public Governance and Sustainable Development: Evidence from 2350 Data” (ID: sustainability-2378110). Those comments are all valuable and very helpful for revising and improving our paper, as well as the important guiding significance to our researches. We have studied comments carefully and have made correction which we hope meet with approval. Due to the heavy workload of revision, we have tried our best to revise our manuscript according to the comments. Attached please find the revised version, which we would like to submit for your kind consideration.

The main corrections in the paper and the responds to the reviewer’s comments are as flowing:

Reviewer #4:

  1. Response to comment: ( Such as, The hypothesis of the paper has not been fully revealed.Despite 2350 literature review, 39 references (39th and 40th references are the same) were used. These references generally belong to the regions of China and Africa.However, the authors stated that there has been an extremely rapid increase in the number of publications in the literature since 2015, and the Core research power is mainly distributed in the Chinese Academy of Sciences, Wageningen University and Research and CGIAR. They said that the countries with superior influence status are mainly UK, USA, GERMANY, AUSTRALIA and ITALY. As a result, the study could not go beyond a situation determination. Authors should explain exactly what they want to reveal with this study. Also, what should be done about the subject of the paper, what is the problem or what is missing? The results obtained in the paper should be discussed more concretely. )

Response:

We are grateful to the Reviewer’s comments. Considering the Reviewer’s suggestion, firstly, we have improved the research goal, and supplemented the research questions in this paper, and see red font content on page 3-4 for more details. Secondly, in the first submitted manuscript, although the titles of the 39th and 40th references are the same, the content of the versions are different, and references are mainly labeled according to actual citation needs. Thirdly, we have also made corresponding modifications and improvements to other parts, and see red font content for more details.

For the research goal and research questions:

In light of this, the purpose of this paper is to focus on taking agricultural and rural public governance and sustainable development as the research theme, which based on the WoS core collection database and CiteSpace metrological analysis software, provides an in-depth examination of the spatial-temporal evolution, cutting-edge map, and logical evolution of research on public governance and sustainable development in agriculture and rural areas. Firstly, it can contribute to reveal and reflect the status and progress, the topics and hot spots, as well as the mainstream and fronts, and the trends and vistas of its research field from a systematic, comprehensive, and entirety perspective for research on public governance and sustainable development in agriculture and rural areas. Secondly, it can contribute to fully grasp the logic context of the development history and the frontier trend of hot knowledge for research on public governance and sustainable development in agriculture and rural areas. Thirdly, it can contribute to provide academic perspectives and research foundations for other scholars or decision-makers to engage in related research of agricultural and rural public governance and sustainable development.

This paper is referred to drawing on the concept model of CiteSpace pioneered by Prof. Chaomei Chen [41,42], this model creatively integrates the methods of citation analysis (diachronic) and co-citation analysis (structural), creates the mapping from knowledge base to research frontier, and highlights the discipline basis of citation network map and the technical basis of information space map [42-44]. With the help of version V.6.2.R1 of the CiteSpace that specialized scientific metrological analysis software with relatively complete functions, this paper is expected to analyze the basic situation, research hotspots, frontier detection, and evolution path of agricultural and rural public governance and sustainable development, in order to find out the knowledge of potential associations contained in the research on agricultural rural public governance and sustainable development, and characterize a series of visual knowledge maps, so as to reveal and reflect the multivariate, time-sharing, and dynamic spatial information, as well as its network relationships and mutual influences, and then to interrogate the temporal and spatial variations, frontier probing and evolution paths in the filed of agricultural rural public governance and sustainable development. The conceptual model of CiteSpace is shown in Figure 1.

In this paper, according to the research goal and conceptual model, an application model of CiteSpace for research on public governance and sustainable development in agriculture and rural areas was created. The application model of CiteSpace on agricultural and rural public governance and sustainable development is depicted in Figure 2. Furthermore, this paper attempts to answer the following questions:

(1) When did the field of agricultural and rural public governance and sustainable development start to be studied? Where is the research stronger? What well-known scholars are paying attention to? What is the cooperation network in this field?

(2) What research hotspots have emerged in the field of agricultural and rural public governance and sustainable development? What research topic clusters have been cohered? What research frontiers are presented?

(3) How has the knowledge base and research paradigm in the field of agricultural and rural public governance and sustainable development evolved? In which years did a cluster appear? In which years did the research results of a certain cluster begin to increase or decrease? In which years did iconic literature appear and affect the overall trend of clustering?

For the Conclusions:

This paper based on the research topic of agricultural and rural public governance and sustainable development, with the help of CiteSpace metrological analysis software, a scientific metrological and knowledge map analysis was conducted on spatial-temporal evolution, collaboration network, research hotspots, cluster labels, frontier detection, and evolution path of 2350 data. The main results of this paper are summarized as follows.

(1) Beginning at the end of the 20th century, scholars began to focus on the field of agricultural and rural public governance and sustainable development and conduct research. Since 2015, there has been an extremely rapid increase in the number of publications in the literature. The core research force is mainly distributed in Chinese Academy of Sciences, Wageningen University & Research, and CGIAR. The ten countries/regions with the highest node centrality are ENGLAND, USA, GERMANY, AUSTRALIA, ITALY, CANADA, SLOVAKIA, NETHERLANDS, PEOPLES R CHINA and INDIA. The collaboration network among authors, between institutions, and between countries/regions each had a certain degree of connection in different periods, respectively.

(2) The structure of knowledge map analysis are significant, and the results are highly reliable. The count for much research hotspots such as sustainable development, rural development, agriculture, and other have influenced the development of the entire research process, and have evolved into larger topic cluster groups such as ecosystem service, sustainable agriculture, land consolidation, and agricultural intensification. It evolved into research frontiers and their knowledge foundations in which the following structures are significant and play key roles, such as agriculture, integrated system, smallholder system, brazilian amazon, rural sustainable development, and land consolidation. Smallholder farms, family farms, critical resources in rural, sustainability of agritourism activity, land use transitions, food security and nutrition has burst an emerging trend in the recent three years.

(3) Agriculture (1996), sustainable development (1998), management (1999), and rural development (2000), which occupy an important position in the corresponding cluster and influenced the trend of the whole cluster in corresponding period, respectively. In the recent three years, research clusters that have received continuous attention include rural revitalization, climate change, property rights, multivariate probit, renewable energy, and sustainable development. Moving towards sustainable development is a common and unchanging issue for humanity.

Based on the above findings, although agricultural and rural public governance and sustainable development is increasingly receiving high attention in academic research and practice environments, there are still many challenges in the research field of agricultural and rural public governance and sustainable development, which further development deeply is urgently required. Firstly, scholars from different countries or regions need to deepen their cooperation network constantly, make academic contributions in the research field of agricultural and rural public governance and sustainable development to achieve the goal of global sustainable development, and jointly build and consolidate the academic community with close cooperation. Secondly, at present, there is a certain consensus in the academic circle around the theme research of agricultural and rural public governance and sustainable development, but it is still necessary to focus on the UN 2030 SDGs, combine the countries and regional development needs and reality, carry out further in-depth discussion in the following research fields, such as sustainable agriculture, smallholder system, food production, rural sustainable development, agricultural green total factor productivity, and anti-poverty, and then put forward countermeasures which conform to the mainstream norms and can be used for reference. Thirdly, governance ability and governance level of sustainable development of global agriculture and rural areas urgently need to be improved. Further clarify and refine the topics that need to be studied and the problems that need to be solved, more scientific demonstration and more feasible measures should be adopted to jointly deal with and solve the current awareness problems and practical challenges, and further promote practical development by cohering academic consensus, expand and innovate the governance model from the comprehensive dimensions of economy, politics, society, culture and ecological environment so as to achieve good governance of agricultural and rural.

This paper has some limitations and future research suggestions, which are as follows: Firstly, this paper reveals that research on agricultural and rural public governance and sustainable development of 2350 data from the core collection database of Web of Science, which has high authority and persuasiveness. However, the core database is basically updated every day. In the future, study samples require data sources to be updated to provide analysis results that keep pace. Secondly, the scientific metrological analysis on agricultural and rural public governance and sustainable development is mainly based on published literature data, and the findings are considered to be of high confidence. However, this study lack a combined analysis of practical cases. In the future, we will consider innovation and fusion research with practice cases.

Thanks very much for your kind work and consideration on publication of our paper. On behalf of my co-authors, we would like to express our great appreciation to editor and reviewers.

Thank you and best regards.

Sincerely yours,

Name: Tingting Huang

E-mail: Sherohtt@foxmail.com

Reviewer 5 Report

The authors analyzed the studies on agricultural and rural public governance and sustainable development through application of CiteSpace model. The study is very useful to see the overall research on  agricultural and rural public governance and sustainable development. But the following revisions would increase the significance of the paper.

 There are grammer errors in the paper. Therefore, the language of the paper should be checked.

 The authors clearly explain what the goal and novelty of the study are in the introduction.

 The athors should explain the motivation behind the methods of the study and advantages and disadvantages of the research methods.

 The authors should add the findings of similar studies to the paper.

 The authors explain how they decided the study period?

 The authors should explain the findings of analyses (for example “Since 2015, there has been an extremely rapid increase in the number of publications in the literature”)

 The authors should add the limitations of the study and future research suggestions to the paper.

There are grammar problems in the paper as I see. Therefore, it should be checked before publication.

Author Response

Dear Editors and Reviewers:

Thank you for your letter and for the reviewer’s comments concerning our manuscript entitled “Research on Agricultural and Rural Public Governance and Sustainable Development: Evidence from 2350 Data” (ID: sustainability-2378110). Those comments are all valuable and very helpful for revising and improving our paper, as well as the important guiding significance to our researches. We have studied comments carefully and have made correction which we hope meet with approval. Due to the heavy workload of revision, we have tried our best to revise our manuscript according to the comments. Attached please find the revised version, which we would like to submit for your kind consideration.

The main corrections in the paper and the responds to the reviewer’s comments are as flowing:

Reviewer #5:

  1. Response to comment: ( There are grammer errors in the paper. Therefore, the language of the paper should be checked.)

Response:

Thanks for the Reviewer’s comments. Considering the Reviewer’s suggestion, we have optimized the English language expression of this manuscript, and see red font content for more details. We sincerely hope that the reviewer can provide clear guidance and correction.

  1. Response to comment: ( The authors clearly explain what the goal and novelty of the study are in the introduction.)

Response:

We are grateful to the Reviewer’s comments.

  1. Response to comment: ( The athors should explain the motivation behind the methods of the study and advantages and disadvantages of the research methods.)

Response:

Thanks for the Reviewer’s comments. Considering the Reviewer’s suggestion, we have improved the motivation behind the methods of the study, and see red font content on page 3-4 for more details.

This paper is referred to drawing on the concept model of CiteSpace pioneered by Prof. Chaomei Chen [41,42], this model creatively integrates the methods of citation analysis (diachronic) and co-citation analysis (structural), creates the mapping from knowledge base to research frontier, and highlights the discipline basis of citation network map and the technical basis of information space map [42-44]. With the help of version V.6.2.R1 of the CiteSpace that specialized scientific metrological analysis software with relatively complete functions, this paper is expected to analyze the basic situation, research hotspots, frontier detection, and evolution path of agricultural and rural public governance and sustainable development, in order to find out the knowledge of potential associations contained in the research on agricultural rural public governance and sustainable development, and characterize a series of visual knowledge maps, so as to reveal and reflect the multivariate, time-sharing, and dynamic spatial information, as well as its network relationships and mutual influences, and then to interrogate the temporal and spatial variations, frontier probing and evolution paths in the filed of agricultural rural public governance and sustainable development. The conceptual model of CiteSpace is shown in Figure 1.

In this paper, according to the research goal and conceptual model, an application model of CiteSpace for research on public governance and sustainable development in agriculture and rural areas was created. The application model of CiteSpace on agricultural and rural public governance and sustainable development is depicted in Figure 2. Furthermore, this paper attempts to answer the following questions:

(1) When did the field of agricultural and rural public governance and sustainable development start to be studied? Where is the research stronger? What well-known scholars are paying attention to? What is the cooperation network in this field?

(2) What research hotspots have emerged in the field of agricultural and rural public governance and sustainable development? What research topic clusters have been cohered? What research frontiers are presented?

(3) How has the knowledge base and research paradigm in the field of agricultural and rural public governance and sustainable development evolved? In which years did a cluster appear? In which years did the research results of a certain cluster begin to increase or decrease? In which years did iconic literature appear and affect the overall trend of clustering?

  1. Response to comment: ( The authors should add the findings of similar studies to the paper.)

Response:

 According to the Reviewer’s comments, we have further optimized the literature review, and see red font content on page 2-3 for more details.

At the same time, scholars have combed out a number of conditions that contribute to public governance and sustainable development in agriculture and rural areas, the following aspects are included, reduce deforestation, carry out the intensification management of agricultural regions [4], transform the utilization mode of cropland in rural areas [5], put into effect the grassroots anti-pesticide mobilisation [6], develop agricultural extension activities, support rural credit, improve agricultural mechanisation, expand the marketing of agricultural and rural [7], develop demonstration farms [8-13], cultivate a community for rural environmental governance [14], as well as increase agricultural production efficiency [15]. However, there are also some practical and empirical results that reveal the reasons or factors that hinder public governance and sustainable development in agriculture and rural areas, for instance, agricultural land fragmentation [16], change in climate, technology, policy and market prices [17], lack of leadership and overall planning for sustainable development of resources and environment [18], weak rural education, inadequate labour supply, agricultural extension services are not yet universal, insufficient social capital, risk mitigation attitudes are not optimistic enough, less farming experience and restricted by soil conditions [2]. In conclusion, there are many factors, which can be summed up as follows, agricultural economy, agricultural productivity, farm size, market access, agroecological potential, agricultural product supply chain, rural industrial, work environment, live conditions, infrastructure, public services, public involvement, rural culture, government-related departments, educational resources, health and welfare, social governance, natural, physical, environment, financial, and social capitals, as well as corporate social responsibility [19-24]. For all these reasons, it is necessary to improve the economic, social and environmental influence of agricultural and rural areas as a whole [25], government and non-governmental organizations improve the rural and agricultural development policies [26], define policies that are socially and environmentally acceptable and geared to tackling the complex challenges [27], advance the development of plans and strategies for sustainable development of villages [28], priority should be given to the construction of transportation infrastructure, regulation of farmland transfer, industrial integration, promotion of rural entrepreneurship, and land consolidation [29], improve public investments in infrastructures, human capital and technology in agriculture and rural area to enhance the competitiveness [30], strengthen more effective public forest governance [4], develop multi-talent rural education and integrating first-second-third industries [31-33], facilitate bridging the technical and associative potential of the agroecological productive [34], increase farmers’ income, form a more complete agricultural product supply chain, and highlight the agricultural brand effect [35,36]. Simultaneously, pay more attention to the reconstruction of governance structure and governance model in rural [37], explore and towards developing a geoscientific approach to public governance and sustainable development in agriculture and rural areas [38], an in-depth analysis of the internal mechanism of the evolution of agricultural production patterns at different phases [39], and give impetus to the multifunctional rural development [40]. From this, it can be seen that, to achieve public governance and sustainable development in agriculture and rural areas, the improvement of both agricultural economic environment and rural development environment should be considered [21].

To summarize, there is no denying the fact that these academic literature provide a noteworthy reference for decision-makers in their follow-up public governance and sustainable development planning in agriculture and rural areas. However, objectively speaking, although there are a large number of existing researches on public governance and sustainable development in agriculture and rural areas, there are indeed rare to focus on the dynamic progress, hot spot analysis, frontier detection, evolution logic and trend outlook, and the scientific metrological analysis of previous research results is even rare, which are not conducive to other scholars or readers recognizing the importance, authority and representativenesss of research results from the massive amount of literature information data. In light of this, the purpose of this paper is to focus on taking agricultural and rural public governance and sustainable development as the research theme, which based on the WoS core collection database and CiteSpace metrological analysis software, provides an in-depth examination of the spatial-temporal evolution, cutting-edge map, and logical evolution of research on public governance and sustainable development in agriculture and rural areas. Firstly, it can contribute to reveal and reflect the status and progress, the topics and hot spots, as well as the mainstream and fronts, and the trends and vistas of its research field from a systematic, comprehensive, and entirety perspective for research on public governance and sustainable development in agriculture and rural areas. Secondly, it can contribute to fully grasp the logic context of the development history and the frontier trend of hot knowledge for research on public governance and sustainable development in agriculture and rural areas. Thirdly, it can contribute to provide academic perspectives and research foundations for other scholars or decision-makers to engage in related research of agricultural and rural public governance and sustainable development.

  1. Response to comment: ( The authors explain how they decided the study period?)

Response:

Thanks for the Reviewer’s comments. The study period was not our decision, but was derived from data sources.

  1. Response to comment: ( The authors should explain the findings of analyses (for example “Since 2015, there has been an extremely rapid increase in the number of publications in the literature”))

Response:

Thanks for the Reviewer’s comments. Considering the Reviewer’s suggestion, we have revised and improved this part of “3.1. Trends in the Number of Published Papers”, and see red font content on page 7 for more details.

Figure 3 depicts the trend of published papers on agricultural and rural public governance and sustainable development, which helps to intuitively grasp the publication dynamics of research on public governance and sustainable development in agriculture and rural areas. From the overall characteristics, since 1996, the core literature of agricultural and rural public governance and sustainable development research has appeared, the fluctuation in the number of publications in the literature on agricultural and rural public governance and sustainable development is very flat between 1996 and 2015, with no more than 70 publications in the highest annual volume. However, it is surprising that from 2016 to 2022, the number of literature publications showed a rapid linear increase. During this period, the lowest annual publication volume was 81 in 2016 and reached the highest annual publication volume of 420 in 2022. This means that after 2015, the number of publications of research field in the agricultural and rural public governance and sustainable development showes a steep slope growth, which has a great relationship with the UN summit on Sustainable Development in 2015 adopted 17 sustainable development goals with the aim of turning to the path of sustainable development, and has also caused in-depth research and extensive discussion on this theme of the academic community.

  1. Response to comment: ( The authors should add the limitations of the study and future research suggestions to the paper.)

According to the Reviewer’s comments, we have added the limitations of the study and future research suggestions to the paper, and see red font content on page 20 for more details.

This paper has some limitations and future research suggestions, which are as follows: Firstly, this paper reveals that research on agricultural and rural public governance and sustainable development of 2350 data from the core collection database of Web of Science, which has high authority and persuasiveness. However, the core database is basically updated every day. In the future, study samples require data sources to be updated to provide analysis results that keep pace. Secondly, the scientific metrological analysis on agricultural and rural public governance and sustainable development is mainly based on published literature data, and the findings are considered to be of high confidence. However, this study lack a combined analysis of practical cases. In the future, we will consider innovation and fusion research with practice cases.

Thanks very much for your kind work and consideration on publication of our paper. On behalf of my co-authors, we would like to express our great appreciation to editor and reviewers.

Thank you and best regards.

Sincerely yours,

Name: Tingting Huang

E-mail: Sherohtt@foxmail.com

Round 2

Reviewer 1 Report

Hoping to have contributed with my comments to make the paper more effective, I would like to thank the Authors for taking these into account in the review of the article.

Author Response

Dear Editors and Reviewers:

Thank you for your letter and for the reviewer’s comments concerning our manuscript entitled “Research on Agricultural and Rural Public Governance and Sustainable Development: Evidence from 2350 Data” (ID: sustainability-2378110). Those comments are all valuable and very helpful for improving our paper, as well as the important guiding significance to our researches. We have studied comments carefully and have made correction which we hope meet with approval. We have tried our best to revise our manuscript according to the comments. Attached please find the revised version, which we would like to submit for your kind consideration.

The main corrections in the paper and the responds to the reviewer’s comments are as flowing:

Reviewer #1:

  1. Response to comment: ( Hoping to have contributed with my comments to make the paper more effective, I would like to thank the Authors for taking these into account in the review of the article. )

Response:

We are grateful to the Reviewer’s comments. Thank you very much for your acknowledgement of the improvement of our manuscript and for agreeing to accept our paper.

Thanks very much for your kind work and consideration on publication of our paper. On behalf of my co-authors, we would like to express our great appreciation to editor and reviewers.

Thank you and best regards.

Sincerely yours,

Name: Tingting Huang

E-mail: Sherohtt@foxmail.com

Reviewer 2 Report

This manuscript has been revised sufficiently. It can be accepted.

Author Response

Dear Editors and Reviewers:

Thank you for your letter and for the reviewer’s comments concerning our manuscript entitled “Research on Agricultural and Rural Public Governance and Sustainable Development: Evidence from 2350 Data” (ID: sustainability-2378110). Those comments are all valuable and very helpful for improving our paper, as well as the important guiding significance to our researches. We have studied comments carefully and have made correction which we hope meet with approval. We have tried our best to revise our manuscript according to the comments. Attached please find the revised version, which we would like to submit for your kind consideration.

The main corrections in the paper and the responds to the reviewer’s comments are as flowing:

Reviewer #2:

  1. Response to comment: ( This manuscript has been revised sufficiently. It can be accepted.)

Response:

We are grateful to the Reviewer’s comments. Thank you very much for your acknowledgement of the improvement of our manuscript and for agreeing to accept our paper.

Thanks very much for your kind work and consideration on publication of our paper. On behalf of my co-authors, we would like to express our great appreciation to editor and reviewers.

Thank you and best regards.

Sincerely yours,

Name: Tingting Huang

E-mail: Sherohtt@foxmail.com

Reviewer 3 Report

Dear authors, thank you for submitting an improved manuscript. Research has indeed become more faithful and scientific. However, I would still like to see the research hypothesis when answering comment 1. When answering comment 4, it is indicated what needs to be done, but it is not indicated how, using what methods, tools, mechanisms. Please write this point in more detail.

Thank you for improving the quality of English.

Author Response

Dear Editors and Reviewers:

Thank you for your letter and for the reviewer’s comments concerning our manuscript entitled “Research on Agricultural and Rural Public Governance and Sustainable Development: Evidence from 2350 Data” (ID: sustainability-2378110). Those comments are all valuable and very helpful for improving our paper, as well as the important guiding significance to our researches. We have studied comments carefully and have made correction which we hope meet with approval. We have tried our best to revise our manuscript according to the comments. Attached please find the revised version, which we would like to submit for your kind consideration.

The main corrections in the paper and the responds to the reviewer’s comments are as flowing:

Reviewer #3:

  1. Response to comment: ( thank you for submitting an improved manuscript. Research has indeed become more faithful and scientific. However, I would still like to see the research hypothesis when answering comment 1. When answering comment 4, it is indicated what needs to be done, but it is not indicated how, using what methods, tools, mechanisms. Please write this point in more detail.)

Response:

Thanks for the Reviewer’s comments. Considering the Reviewer’s comments, we have supplemented the research hypothesis in this paper, and see blue font content on page 4 for more details. At the same time, we have optimized the recommendation of the study, and see blue font content on page 20 for more details. Thanks very much for your kind work and consideration on publication of our paper.

Taking the above analyses together, combined with the practical aspects of the research content of this paper, based on which attempt to propose theoretical hypotheses are as follows.

Hypothesis 1. Scholars from different countries or regions have a certain degree of cooperation in the research field of agricultural and rural public governance and sustainable development, but the network relationship of academic cooperation is not very strong.

Hypothesis 2. The research hotspots in the research field of agricultural and rural public governance and sustainable development embody the SDGs from the macro level, but the specific research hotspots in different periods are uncertain.

Hypothesis 3. The evolution path of the knowledge base and research paradigm in the research field of agricultural and rural public governance and sustainable development  are uncertain.

Based on the above findings, although agricultural and rural public governance and sustainable development is increasingly receiving high attention in academic research and practice environments, there are still many challenges in the research field of agricultural and rural public governance and sustainable development, which further development deeply is urgently required. Firstly, deepen academic exchanges and strengthen academic cooperation. In this paper, the map of the author cooperative network show that consists of 805 nodes and 502 lines, and the density of the co-occurrence network is 0.0016, which the cooperation intensity needs to be strengthened. Therefore, scholars from different countries or regions need to deepen their cooperation network constantly, make academic contributions in the research field of agricultural and rural public governance and sustainable development to achieve the goal of global sustainable development, and jointly build and consolidate the academic community with close cooperation. Secondly, promote in-depth research on important issues in the field of agricultural and rural public governance and sustainable development. In this paper, keywords co-occurrence analysis results show that the three keywords with the highest centrality are sustainable development, rural development and agriculture. Terms cluster knowledge analysis results show that the three terms cluster with the highest label value are ecosystem service, sustainable agriculture and land consolidation. Reference co-citation analysis show that the three terms cluster with the highest label value (LLR) are agriculture, integrated system and smallholder system. The strongest sigma and burstness analysis results show that smallholder farms, family farms, rural critical resources, agritourism activity, land use and management, food security and nutrition have burst and become an emerging trend in the recent three years. At present, there is a certain consensus in the academic circle around the theme research of agricultural and rural public governance and sustainable development, but it is still necessary to focus on the UN 2030 SDGs, combine the countries and regional development needs and reality, carry out further in-depth discussion in the following research fields, such as sustainable agriculture, smallholder system, food production and security, land consolidation, rural sustainable development, and agricultural green total factor productivity [45], and then put forward countermeasures which conform to the mainstream norms and can be used for reference. Thirdly, strengthen public governance in agriculture and rural areas. In this paper, the evolution path analysis results show that ten clusters and their keywords, which actually reflect the public governance issues that have important impacts and are of great concern in the process of agricultural and rural sustainable development. Therefore, governance ability and governance level of sustainable development of global agriculture and rural areas urgently need to be improved. Further clarify and refine the topics that need to be studied and the problems that need to be solved, more scientific demonstration and more feasible measures should be adopted to jointly deal with and solve the current awareness problems and practical challenges, and further promote practical development by cohering academic consensus, expand and innovate the governance model from the comprehensive dimensions of economy, politics, society, culture and ecological environment so as to achieve good governance of agricultural and rural.

Thanks very much for your kind work and consideration on publication of our paper. On behalf of my co-authors, we would like to express our great appreciation to editor and reviewers.

Thank you and best regards.

Sincerely yours,

Name: Tingting Huang

E-mail: Sherohtt@foxmail.com

Reviewer 5 Report

Dear Authors;

You considerably improve the paper based on the suggestions.

I suggest "accept in present form."

The language of the paper seems good after the revision.

Author Response

Dear Editors and Reviewers:

Thank you for your letter and for the reviewer’s comments concerning our manuscript entitled “Research on Agricultural and Rural Public Governance and Sustainable Development: Evidence from 2350 Data” (ID: sustainability-2378110). Those comments are all valuable and very helpful for improving our paper, as well as the important guiding significance to our researches. We have studied comments carefully and have made correction which we hope meet with approval. We have tried our best to revise our manuscript according to the comments. Attached please find the revised version, which we would like to submit for your kind consideration.

The main corrections in the paper and the responds to the reviewer’s comments are as flowing:

Reviewer #5:

  1. Response to comment: ( You considerably improve the paper based on the suggestions. I suggest accept in present form.)

Response:

We are grateful to the Reviewer’s comments. Thank you very much for your acknowledgement of the improvement of our manuscript and for agreeing to accept our paper.

Thanks very much for your kind work and consideration on publication of our paper. On behalf of my co-authors, we would like to express our great appreciation to editor and reviewers.

Thank you and best regards.

Sincerely yours,

Name: Tingting Huang

E-mail: Sherohtt@foxmail.com